# Insights into Biodegradable Polymer-Supported Titanium Dioxide Photocatalysts for Environmental Remediation

**Nina Maria Ainali** [1,2], **Dimitrios Kalaronis** [2], **Eleni Evgenidou** [2,3], **Dimitrios N. Bikiaris** [1] and **Dimitra A. Lambropoulou** [2,3,*]

1. Laboratory of Polymer Chemistry and Technology, Department of Chemistry, Aristotle University of Thessaloniki, GR-54124 Thessaloniki, Greece; nsainali@chem.auth.gr (N.M.A.); dbic@chem.auth.gr (D.N.B.)
2. Laboratory of Environmental Pollution Control, Department of Chemistry, Aristotle University of Thessaloniki, GR-54124 Thessaloniki, Greece; dkalaron@chem.auth.gr (D.K.); evgenido@chem.auth.gr (E.E.)
3. Center for Interdisciplinary Research and Innovation (CIRI-AUTH), Balkan Center, GR-57001 Thessaloniki, Greece
* Correspondence: dlambro@chem.auth.gr; Tel.: +30-2310-997-687; Fax: +30-2310-997-859

**Abstract:** During the past two decades, immobilization of titanium dioxide (TiO$_2$), a well-known photocatalyst, on several polymeric substrates has extensively gained ground since it limits the need of post-treatment separation stages. Taking into account the numerous substrates tested for supporting TiO$_2$ photocatalysts, the use of biodegradable polymer seems a hopeful option owing to its considerable merits, including the flexible nature, low price, chemical inertness, mechanical stability and wide feasibility. The present review places its emphasis on recently published research articles (2011–2021) and exhibits the most innovative studies facilitating the eco-friendly biodegradable polymers to fabricate polymer-based photocatalysts, while the preparation details, photocatalytic performance and reuse of the TiO$_2$/polymer photocatalysts is also debated. The biodegradable polymers examined herein comprise of chitosan (CS), cellulose, alginate, starch, poly(lactid acid) (PLA), polycaprolactone (PCL) and poly(lactide-co-glycolide) (PLGA), while an emphasis on the synthetical pathway (dip-coating, electrospinning, etc.) of the photocatalysts is provided.

**Keywords:** biodegradable polymers; titanium dioxide; immobilization; photocatalysis; wastewater remediation

## 1. Introduction

Water contamination by organic compounds and metals has been outlined as one of the major global problems nowadays. In fact, due to their non-biodegradable nature, these harmful compounds remain for long time periods after their discharge into the environment, thus being characterized as persistent contaminants. Several techniques have been explored to remove these pollutants from water, including adsorption and photocatalysis, which comprise attractive and eco-friendly approaches. The nano-sized titanium dioxide (TiO$_2$) is a famous photocatalyst among the metal oxides, due to its excellent efficiency, low price, physicochemical stability, extensive disposal, safety, and non-corrosive behavior. It has three crystal forms, anatase, rutile and brookite, while the first presents the most effective photocatalytic performance. Nevertheless, due to the challenges that arise from the very small particle size and the unfeasible reusability of the particles, including the post separation and recovery of the photocatalytic particles after water or wastewater treatment, the need of TiO$_2$ immobilization is crucial [1].

In this context, several polymers have been proposed as supporting materials to immobilize the inorganic TiO$_2$ nanoparticles and provide composite materials with advanced photocatalytic performance [2–4]. Properties such as chemical inertness, mechanical durability, high UV-resistance, wide density range (0.9–2 g cm$^{-1}$), low-cost and extensive availability render polymeric materials as the optimal ones for photocatalytic processes.

However, the wide overuse of petroleum-derived polymeric materials in the membrane waste remediation systems is an aspect with adverse effects on the environment. Biodegradable polymeric materials have been recently employed as green alternatives to replace the already used petroleum-based counterparts and minimize by this way their harmful fingerprint [5].

This paper provides a guiding strategy for the selection of optimal approaches to the fabrication of biodegradable polymers that are suitable for the manufacture of $TiO_2$-immobilized photocatalysts with enhanced photocatalytic performance towards wastewater remediation. Therefore, this short review survey aims to outline and discuss the most representative, innovative and current advancements (2011–2021) in the field of the fabrication of $TiO_2$-immobilized materials supported on biodegradable polymers, both natural or synthetic, including chitosan (CS), cellulose, alginate, starch for the former category and poly(lactic acid) (PLA), polycaprolactone (PCL) and poly(lactide-*co*-glycolide) (PLGA) for the latter. The photocatalytic performance of the enclosed $TiO_2$-composite materials against several organic compounds, dyes, metals, and pharmaceuticals is also presented in brief. Considering the challenge faced by modern societies and the environmental pollution, we highlight the demand of exploring new and eco-friendly photocatalysts and investigate their performance against various pollutants. A schematic depiction of the topics enclosed within the present review is illustrated at Figure 1.

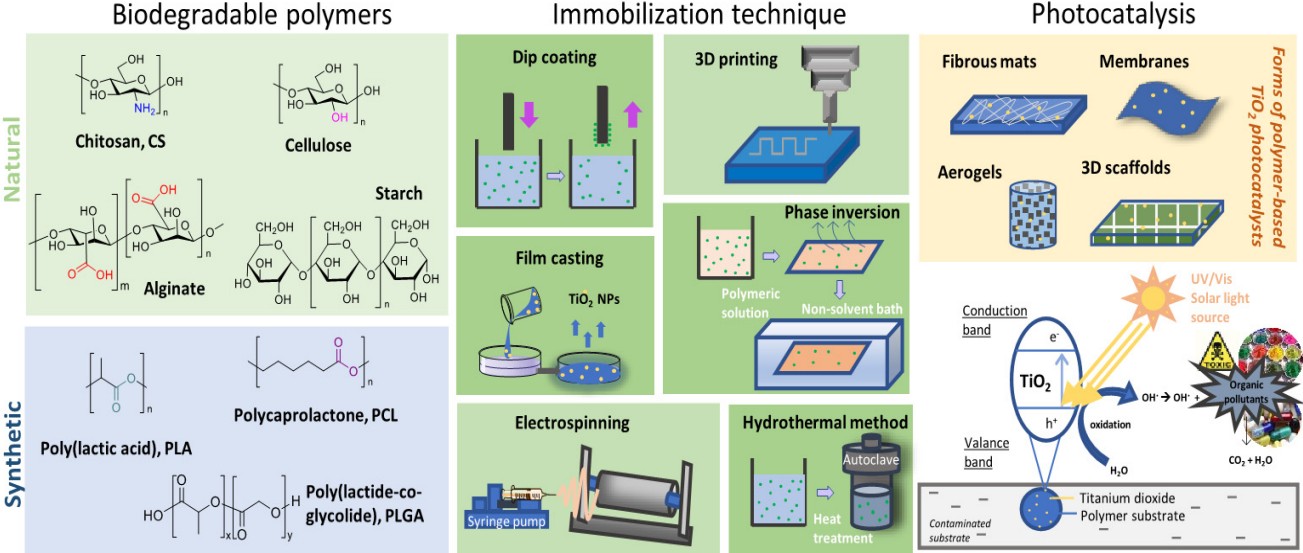

**Figure 1.** A brief illustration of the main synthetic routes fabricated for the immobilization of $TiO_2$ nanoparticles onto biodegradable polymeric substrates for the photocatalytic degradation of organic pollutants.

## 2. Biodegradable Polymers Combined with $TiO_2$ for Enhanced Photocatalytic Activity

Many recent studies have been focused on using biodegradable polymers as templates or supporting materials for the direct anchoring of $TiO_2$ particles and for the manufacturing of photocatalysts. These polymers seem like ideal substitutes for the non-biodegradable polymeric substrates, due to their environmentally friendly nature and the obvious merits of their use. Both natural and synthetic polymers can possess a biodegradable nature; however, most work has been conducted employing the former category. Despite the dynamic and promising character of natural polymers towards the synthesis of effective $TiO_2$-photocatalysts, in many studies, the fabrication of a synthetic non-biodegradable polymer in a combination way seems vital in order to construct stable and more durable structures. However, limited attempts were made to combine natural and synthetic biodegradable polymers in this field, as reported in few studies [6,7].

The method chosen for the immobilization of $TiO_2$ on the support greatly affects the photocatalytic performance of titania and thus should be carefully selected, based

on the employed substrate and the targeted pollutant [2]. Generally, it is advisable that the selected depositing method does not provoke a deterioration of the photocatalytic activity of $TiO_2$. Several methods have been proposed in the literature, including sol-gel method [8] consisting primarily of film casting [4] and dip-coating methods [9], hydrothermal treatment [10], plasma treatment [11], electrospinning [12], electrospraying [13] and 3D printing [14]. The sol-gel method for the nano-sized $TiO_2$ particles is based on the hydrolysis of titanium alkoxides (Ti(O-E), Ti(i-OP)$_4$ or Ti(O-nBu)$_4$) at room temperature. Due to its simple and low-cost character, this pathway is mostly preferred. Nevertheless, since an efficient photocatalytic performance, triggered by the photo-induced holes and electrons' generation, is favored in well-crystallized phases which require high calcination temperatures (commonly 300 °C), the need of supporting-polymeric materials of high thermal stability is urgent. The latter limits the variety of used polymers and thus further exploration of sol-gel in this field is advised.

Concerning the application of the photocatalyst, different experimental conditions can be employed, while its performance can be tested in a primary adsorption step as well. For generating OH$^\bullet$ radicals during photocatalytic reaction, various light sources are selected emitting at different wavelengths (UV/visible/sunlight) or intensities. Photocatalysis can be affected by various factors such as initial concentration of contaminant, pH, temperature, as well as the structure of materials. Increase of the targeted pollutant's initial concertation can affect its degradation efficiency since less available hydroxyl radicals have to degrade a larger amount of organic matter, while the concentration of catalyst remains stable [15]. The pH is chosen according to the contaminant, and it can range between acidic and basic values. Some compounds can be degraded effectively in acidic environment, near to neutral pH, or in basic pH [16–18]. The effect of temperature can also be a significant factor in the removal efficiency of pollutants. Increase of temperature can improve the degradation efficiency however, very high increase can become inhibitory since it can affect the surface of polymeric material, mainly its crystallinity, and reduce its photocatalytic efficiency. Furthermore, the surface area is strongly affected by the immobilization technique which is observed in conventional fabricated polymeric materials combined with $TiO_2$. In these cases, $TiO_2$ is immobilized into the polymeric matrix, while the photocatalytic activity presents a deterioration tendency in comparison with photocatalytic activity of $TiO_2$ added in suspension in the aqueous solution, and thus, further investigation of the established methodologies should be performed [19].

In our literature survey, which was conducted for the last decade, the most representative examples in the field of photocatalytic biodegradable polymer-$TiO_2$ composite materials are selectively picked, in terms of their innovation and promising character towards the photodegradation of several pollutants, and a short discussion is supported herein.

*2.1. Natural Biodegradable Polymers*

2.1.1. Chitosan (CS)

Synthetic and Characterization Routes

A biodegradable polymer that has been widely explored in green pathways for waste remediation and photocatalytic activities is chitosan (CS). It is a linear polysaccharide and one of the most abundant biopolymers in the nature, with biodegradable, biocompatible and non-toxic character, derived from the deacetylation process of chitin, found in the exoskeletons of crustaceans and arthropods. Enzymes, such as chitosanase or lysozymes, are known to degrade chitosan. Its low-cost and the several versatile properties that chitosan possesses, render this polymer as an ideal candidate for environmental remediation purposes. Since CS is a great supporting material for the dispersion of $TiO_2$ nanoparticles, $CS/TiO_2$ is one of the most investigated composite photocatalysts, while their synergistic effects between them have also been studied. The immobilization of $TiO_2$ in CS films has been widely investigated since it can be easily obtained owing to the miscibility between CS and hydrophilic $TiO_2$. Chitosan contains in its structure amino and hydroxyl

functional groups which act as coordination sites to form complexes with metals and several compounds, boosting by this means the effective removal of pollutants with special selectivity. However, the immobilization attempts require strong affinity between the $TiO_2$ and the substrate, and thus, cross-linking processes with the aid of several alkaline agents (e.g., NaOH) are often selected [15]. A brief description of the studies reported herein for CS-supported photocatalysts is presented in Table 1.

In other cases, the inclusion of organic or inorganic nanofillers, such as carbon dots (CDs) [20], graphene oxide (GO) [21], carbon nanotubes [22] and other metal oxides [18], into the polymer matrix has been also explored, providing nanocomposite materials with excellent architecture and dispersion of the nanofillers. Carbonaceous and inorganic materials not only perform as supporting materials for the new composites but also act as co-catalysts for the intensification of the photocatalytic efficiency of $TiO_2$ [21]. In an attempt to reduce the swelling and improve the physicochemical characteristics and stability of CS, Bahrudin et al. [23] incorporated montmorillonite (Mt) clays into the CS polymeric matrix. Due to the strong hydrogen interactions between the hydroxyl functional groups of CS with the Si–O–Si units of Mt, CS–Mt composite developed into an adsorbent sub-layer for the immobilization of $TiO_2$. The synthetic pathway of these $TiO_2$/CS-Mt bilayer photocatalysts encompassed three individual stages, including: the preparation of the CS-Mt adsorbent plates via casting method, the $TiO_2$ formulation and, finally, the coating of the CS–Mt plate with the $TiO_2$ formulation.

Commonly, the immobilization of the nanofiller into the chitosan matrix is attained by its in situ deposition on the polymer surface. In this context, Midya et al. [20] reported on the in situ deposition of $TiO_2$ nanoparticles and CDs onto the surface of polyvinyl imidazole crosslinked CS, with the aid of microwave irradiation. Crosslinked CS was formed via a radical polymerization using diurethane dimethacrylate (DUDMA) as crosslinker at the propagation stage during the reaction of CS with 1-vinylimidazole (VI), under $N_2$ atmosphere and microwave irradiation. After the formation of the gel, titanium isopropoxide solution and sugar cane juice was added in the reaction system, and the irradiation continued at 75 °C. This in situ methodology, fabricating green CDs from sugar cane juice under microwave assistance was employed for the first time, and the results indicated the effective deposition of titania and CDs onto the layer of the crosslinked CS. Another approach taking advantage of the microwave assistance was reported by Tian et al. [18]. Herein, the innovative part was the formation of the $TiO_2$/$ZrO_2$ composites by a microwave solvothermal method, while the carboxymethyl product of CS was chosen as the polymeric supporting material, due to its higher hydrophilicity, stability and content of functional groups, as well as larger specific area, characteristics of great importance for the coordination with metal oxides and dyes.

Three-dimensional printing is gaining ground lately as a smart technique in materials science, since it allows the formation of identical objects, defined in terms of intended geometries rather than dimensions that can be randomly determined throughout the design stage. Within this concept, Bergamonti et al. [14] utilized a 3D printer to prepare CS scaffolds as the embedding matrix for $TiO_2$ nanoparticles. In order to prepare a stable polymeric network, the dried scaffolds after the 3D printing procedure were further proceeded with ammonia solution in order to eliminate the acetic acid used for the solution of CS and provide by this ionic gelation path a neutral and stable network.

Due to its versatile and easily applicable nature, electrospinning has gained also great interest for the preparation of nanostructured composite fibers for photocatalytic activities. With an innovative point of view, ZabihiSahebi et al. [22] reported on the formation of cellulose acetate/chitosan/single walled carbon nanotubes/ferrite/titanium dioxide (CA/chitosan/SWCNT/$Fe_3O_4$/$TiO_2$) nanofibers for the removal of metals and dyes. Herein, the combination of these two polymers had a dynamic impact on the efficiency of the final matrix. Despite the numerous merits of CS, it possesses a low mechanical strength, and thus CA was selected to boost the mechanical performance of the first. This enhancement was moreover intended by the addition of the inorganic $Fe_3O_4$ and car-

bonaceous SWCNTs with great surface area and microporous design. Concerning the electrospinning procedure, a blend of CS/CA (60/40 w/w) was dissolved in a mixture of trifluoroacetic/dichloromethane solvents and the SWCNT/Fe$_3$O$_4$/TiO$_2$ composites were dispersed in the former solution with the aid of sonication. Electrospinning optimal conditions were as following: applied voltage 20 kV, tip to collector distance of 10 cm and a flow rate of 0.25 mL/h. Results showed that with the rising loading of the SWCNT/Fe$_3$O$_4$/TiO$_2$ into the polymeric fibrous matrix, the diameters of the fibers (275, 295 and 340 nm for 5, 10 and 15 wt% of composites, respectively) enlarged greatly, too. Thermogravimetric analysis (TGA) indicated also the superior thermal stability of the final electrospun composites in relation to CS/CA thermal properties, mainly due to the incorporation of the carbon nanotubes into the fibers. An additional study taking advantage of the electrospinning technique with CS as polymer nanofiber mat was conducted by Wang et al. [24] (Figure 2). In this work, ultrafine CS membranes doped with TiO$_2$/Ag, were loaded with green algae cells and the synergistic photocatalytic activity of the designed mats was investigated for the removal of Cr(VI). According to the authors, the inorganic addition enhanced the conductivity of the polymer solution and though the electrospinning efficiency of CS solution was improved. TEM images implied that the Ag in situ formed nanoparticles were homogeneously dispersed into the polymer matrix, whereas TiO$_2$ deposition was additionally assured by X-ray spectroscopy analysis. SEM images provided in the study illustrate the photo-degradation of the algae-cells on the surface of the prepared mats.

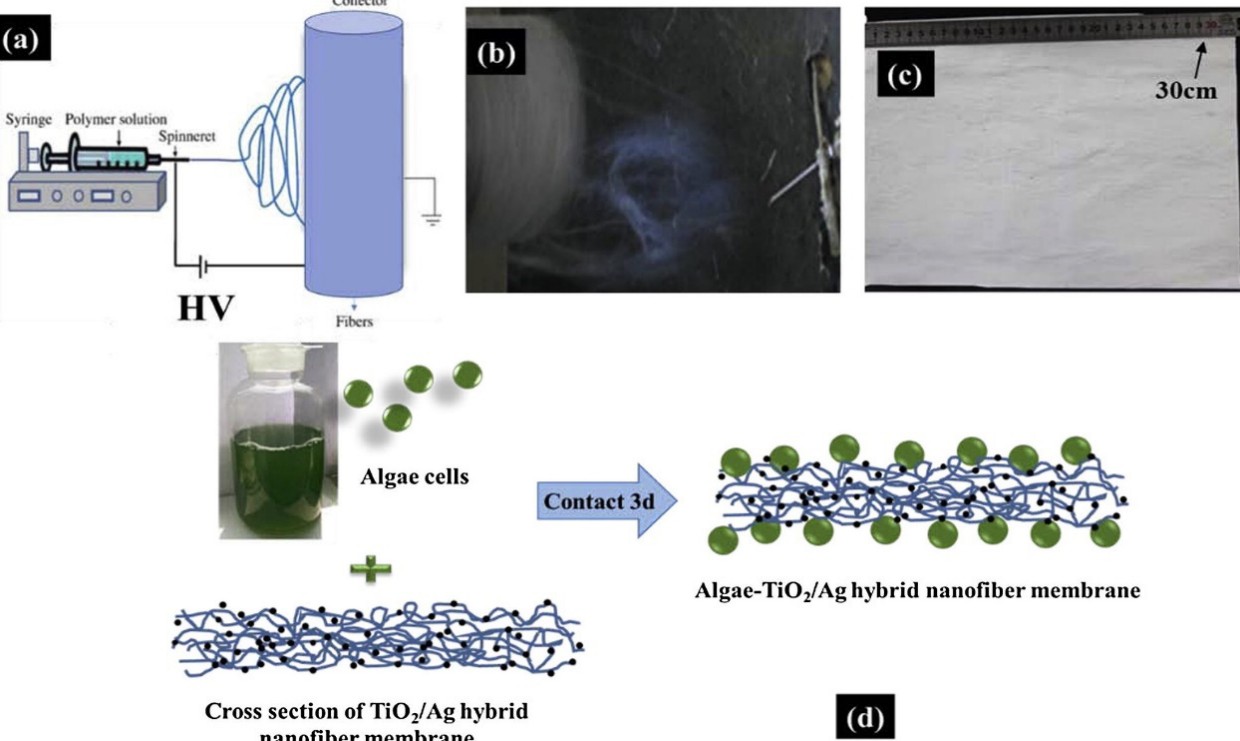

**Figure 2.** (**a**) Schematic illustration of electrospinning system, (**b**) nanofibers generation by single needle electrospinning, (**c**) demonstration of large-scale nanofibers mat and (**d**) scheme modeling the preparation of algae decorated TiO$_2$/Ag hybrid nanofiber membrane. Reprinted with permission from [24].

**Table 1.** Summary of the chitosan-supported TiO$_2$ photocatalysts enclosed in the presented literature.

| | Biodegradable Polymeric Matrix | | | | | Photocatalysis Parameters. | | | Ref. |
|---|---|---|---|---|---|---|---|---|---|
| No. | Polymer Substrate | TiO$_2$ Precursor | Dopant | Immobilization Technique | Morphology of the Photocatalyst | Type of (Target) Pollutant | Light Source | Degradation Efficiency (Time Required) | |
| 1 | CS | TiO$_2$ nanopowders (Aeroxide; 80% anatase) | MT | CS-MT film casting & dip-coating in TiO$_2$ formulation | Bilayer photocatalyst | Methyl orange dye | 45 W fluorescent lamp | 98.7% | [23] |
| 2 | CS-grafted poly(vinyl imidazole) | Titanium isopropoxide | CDs | In situ deposition of TiO$_2$ NPs and CDs onto the polymeric surface under microwave irradiation | Nanocomposite hydrogel | 2,4-dicholorophenol Reactive Blue 4 Reactive Red 15 | Sunlight exposure for 30 min | 95% (180 min) 95.8% (30 min) 98.2% (30 min) | [20] |
| 3 | CS | TiO$_2$ (P25) | - | Immobilization of TiO$_2$ in CS film by cross-linking process | Film | Tetracycline hydrochloride | UV lamp 30 W and λ = 360 nm | 87% (240 min) | [15] |
| 4 | CS | Aeroxide P25-TiO$_2$ | - | 3D printing | 3D printed scaffolds | Amoxicillin | UV irradiation (125 W), λ = 300–800 nm | 90–60% (180 min) | [14] |
| 5 | CS | TiO$_2$ powder (P25) | - | One-step spray-drying synthesis | CS/TiO$_2$ nanocomposite particles | Organic dye, crystal violet | RPR-200 Photochemical Reactor (Rayonet), λ = 300 nm (8×, 21 W), and λ = 350 nm (8×, 24 W) lamps | 58.3–15.5% (120 min) 95.7% pristine particles | [25] |
| 6 | CS | TiO$_2$ | GO | Dopped-GO and CS impregnated in TiO$_2$ solution | - | cefixime trihydrate | 4 × lamps UV-A irradiation, λ = 365 nm | 95.34% (60 min) | [21] |
| 7 | CMCS | Butyl titanate | TiO$_2$/ZrO$_2$ composites | ZrO$_2$:TiO$_2$ were synthesized by a microwave hydrothermal method, CMCS as template | Composites | Rhodamine B | Photochemical reactor-UV irradiation (CEL-LPH120), | 90.5–60.6% (60 min) | [18] |
| 8 | CS + PVA | TiO$_2$ (anatase) | Ag | Loading algae cells on the TiO$_2$/Ag CS hybrid nanofiber mat prepared by electrospinning | Algae-TiO$_2$/Ag hybrid nanofiber membrane | Cr(VI) removal | 500 W halogen tungsten lamp, λ > 400 nm | 91–25% (180 min) | [24] |
| 9 | CS + CA | TiO$_2$ nanoparticles | SWCNTs + Fe$_3$O$_4$ | Incorporated inorganics into electrospun nanofibers | Composite nanofibers | Cr(VI), As(V), Methylene blue and Congo red | 4 × UV lamps, 30 W and λ = 365 nm | ~99% (40–60 min) | [22] |

Abbreviations: CS, chitosan; CMCS, carboxymethyl chitosan; PVA, polyvinyl alcohol; MT, montmorillonite; CDs, carbon dots; SWCNTs, single walled carbon nanotubes; GO, graphene oxide.

Photocatalytic Performance

Concerning the application of CS-based $TiO_2$ materials for removal of various pollutants, nine different studies were selected among others according to their novelty and different fabrication techniques. Different pollutants were removed including dyes used in textile industries, such as: Reactive Blue 4, Reactive Red 15, Methyl Orange (MO), Rhodamine B (RhB), Methylene Blue (MB), Crystal Violet and Congo Red. Furthermore, common antibiotics found in significant amounts globally in wastewaters, were degraded under UV and solar irradiation through chitosan-$TiO_2$ materials applied in photocatalytic experiments. For instance, amoxicillin and tetracycline hydroxide were examined with remarkable results on removal efficiency, while toxic chemical compounds such as Cr(VI), As(V), and 2–4 dichlorophenol were treated also effectively [14,15,20,22,23].

Chitosan based composites achieved a wide range of degradation efficiency reaching 90% in most of the cases, while UV radiation was most frequently employed for the degradation of pollutants. Different structures of chitosan-based materials were used in photocatalytic experiments, such as films, bilayers, 3D printed scaffolds and fibers. Another important factor is the reusability of polymeric materials used in photocatalysis. Chitosan based catalysts were applied for 3–6 cycles in the majority of studies with sufficient results. However, a decrease in the photocatalytic activity was recorded after the second cycle in most of studies. Nevertheless, Bahrudin et al. investigated the photodegradation of methyl orange (MO), in which the reusability of $TiO_2$/CS-Mt composite was tested for 10 cycles. According to this study, two chitosan bilayers were manufactured with or without the addition of Mt. The insertion of Mt showed a favorable effect in charge separation of $TiO_2$. Thus, these bilayers performed a higher photocatalytic performance in degradation of MO dye. Furthermore, $TiO_2$/CS-Mt was reused for ten cycles due to the notable synergism of materials, while the photocatalytic efficiency was almost stable even after ten repetitions. It was also observed that the effect of the aforementioned synergism led to an increase of the reusability of CS photocatalytic material in comparison with other studies [23].

Regarding the time required for the photocatalytic degradation of pollutants, the synthesized composite material as well as the nature of pollutants play a crucial role. Specifically, in most of the cases using CS-$TiO_2$ composites, the photocatalytic activity is completed within two or three hours. However, there are contaminants more stable which need additional time to degrade, such as tetracycline hydrochloride, in which case after 240 min of photocatalysis a high percentage of antibiotic was degraded. Tassadit Ikhlef-Taguelmimt et al. observed that one of the most important factors ameliorating the photocatalytic activity was the agitation speed, since with its increase a decrease of the thickness of boundary layer of $TiO_2$/chitosan film is achieved [15]. Moreover, in a notable study, Lipi Midya et al. investigated the degradation of 2,4-dichlorophenol, Reactive Blue 4, and Reactive Red 15 under sunlight exposure. Herein, different manufactured materials were used such as chitosan polyvinyl imidazole CS-p(VI), CS-p(VI)/$TiO_2$ nanoparticles (NPs), CS-p(VI)/carbon dots (CDs) and CS-p(VI)/$TiO_2$NPs-CDs in hydrogels, as it is described above. The pure CS-p(VI) material performed the lower degradation efficiency, whereas it adsorbed a significant amount of dyes concentration. From the other hand, the CS-p(VI)/$TiO_2$NPs-CDs nanocomposite material exhibited the highest photocatalytic activity. This result was explained by the integration of a visible light material such as CDs that is able to absorb the photons in range of visible irradiation and enhance the charge split of electron/hole pair. Additionally, a rise of surface area was provoked leading to the same result [20]. Moreover, researchers monitored the photocatalytic activity of CS/$TiO_2$ scaffolds. In this context, three separate CS/$TiO_2$ scaffolds were manufactured with three, five, and fifteen layers, each time. The fifteen-layered scaffolds performed the poorest photocatalytic activity since the $TiO_2$ nanoparticles were deeply incorporated into the structure of the material and thus they could not be affected by UV irradiation. On the other hand, three-layered scaffolds showed the best photocatalytic results due to their structure and architecture. Specifically, the latter scaffolds were thicker than the others and possessed a higher surface area; factors leading to a superior photocatalytic

performance. According to their reusability, 3D printed TiO$_2$/CS scaffolds showed high mechanical strength and flexibility, and as a result, these materials were presented as dynamic candidates for several repetition cycles of photocatalytic experiments [14].

### 2.1.2. Cellulose
### Synthetic and Characterization Routes

Cellulose is another non-toxic, biocompatible, as well as biodegradable polysaccharide found in great abundance in nature. It comprises a linear chain with multiple hydroxyl groups able to form hydrogen bonds with other oxygen atoms on the nearby polymeric chain. Biodegradation of cellulose is achieved either by enzymatic oxidation, with peroxidase emitted by fungi, or by bacteria. Cellulose and its derivatives, such as carboxymethyl cellulose, cellulose phosphate, and acetate, have been used primarily as reinforcement materials owing to its excellent mechanical, chemical, and biological properties for dyes and metal binding for environmental remediation purposes. A brief presentation of the studies reported herein for cellulose-supported photocatalysts is illustrated in Table 2.

Several techniques were also utilized for the construction of cellulose-based architectures for the immobilization of TiO$_2$ nanoparticles, including mainly sol-gel, phase-inversion, and casting methods. A facile strategy for the design of triple-layered Au–TiO$_2$ nanoparticles immobilized into porous cellulose membranes was developed by Yu et al. [26]. The fabrication of these membranes was achieved by the tape casting method and the suction filtration procedure for the immobilization of the Au and TiO$_2$ nanoparticles. According to the authors, each layer of the three-layered formed membrane possessed a specific role for the photocatalytic part. The cellulose bottom layer fulfilled a triple role, enclosing its filtering, supporting and thermal insulation efficiency. The middle Au-nanoparticles layer was implied to enhance the photoadsorption efficiency of the TiO$_2$ nanoparticles and to facilitate the separation of the electron-hole pair thus ameliorating the photocatalytic performance of the composite material. Concerning the upper layer comprising of TiO$_2$-nanoparticles, it was clearly that it acted as a photocatalytic surface for the degradation of various contaminants (Figure 3).

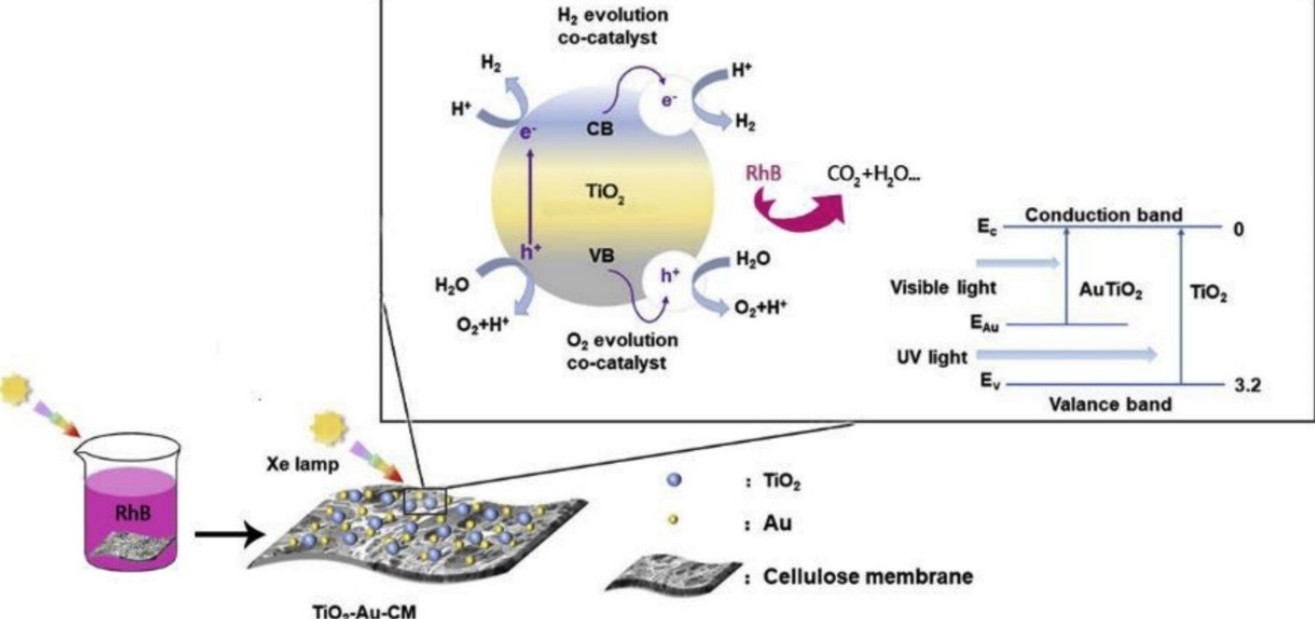

**Figure 3.** Schematic image for the mechanism of photoinduced charge carrier transfers in TiO$_2$-Au cellulose membranes under solar irradiation. Reprinted with permission of Reference [26].

Another promising study utilized regenerated cellulose from recycled newspaper to fabricate a N-doped $TiO_2$ nanorods photocatalytic membrane. According to Mohamed et al. [27], the recycled cellulose/$TiO_2$ nanocomposite films were prepared via a phase inversion technique, whereas FTIR spectroscopy indicated a strong interaction between the abundant –OH groups of the cellulose and the $TiO_2$ nanorods, through hydrogen bonds. Another study utilizing the same technique but the triacetate derivative of cellulose as polymer matrix, was conducted by Taghizadeh et al. [28]. Concerning this work, $TiO_2$ and $TiO_2$/GO were deposited at the surface of the polymeric membranes enhancing though the hydrophilicity of the composite materials, destined for forward osmosis procedure. The application of this method is mainly centered on the desalination of the produced water as well as the simultaneous removal of toxic compounds.

Hierarchical nanocomposite structures have been additionally accepted as dynamic approach for the fabrication of cellulose/$TiO_2$ nanostructured photocatalysts. Lin et al. [29] designed and prepared a hierarchically constructed structure comprising $Ag_2O$-nanoparticle/$TiO_2$-nanotube composites supported by natural cellulose. A sol-gel methodology was applied for the preparation of the 3D-dimensional porous network patterns of cellulose/$TiO_2$ composites, whereas a precipitation method was selected for the synthesis of the final nanocomposites. Results indicated that the enhanced photocatalytic activity of the formed composites was mainly credited to the synergistic and harmonized relation between the titania and silver oxide phases, obtained from the unique architecture.

For the photocatalytic degradation of various cyanotoxins, Pihno et al. [9] designed transparent nine-layered 2D-films of cellulose acetate monoliths coated with a P25 paste or 2D $TiO_2$ prepared by a combination of sol-gel and dip-coating techniques. Firstly, nitric acid, titanium (IV) isopropoxide and water were employed for the preparation of the $TiO_2$ sol-gel, which was further dialyzed in a cellulose membrane with a pH = 3.0. The nine layers were deposited by the dip-coating method with a rate of withdrawal around 0.8 mm s$^{-1}$. After the deposition of each layer, the prepared monoliths were dried (50 °C, 1 h) and further rinsed with deionized water and dried once again (50 °C, 2 h). Few drops of TritonTM X-100 before applying the dip-coating method were added as surface-active agent in order to enhance the adherence of the catalyst layers to the supporting material. SEM images provided in the manuscript reveal that materials' thickness during photocatalytic process remained almost unaffected.

Another study aiming at the preparation of multi-layered films for photocatalytic purposes was conducted by Ullah et al. [30]. According to this work, a flexible layer-by-layer (LbL) procedure for the design of well-dispersed, adherent and porous multi-layered films of $TiO_2$ nanoparticles and phosphotungstic acid (HPW) was based on the utilization of cellulose phosphate (CP) as a polyelectrolyte substate. Herein, the phosphoric ester of cellulose was chosen as the appropriate dispersing agent and chelator toward transition metals, without the need of any special pH-treatment and fast deposition of $TiO_2$ nanoparticles onto the surface of the substrate. Due to the HPW's advanced photochromic and photocatalytic properties, the synergistic photocatalytic efficiency of HPW and $TiO_2$ was expected and finally achieved. Concerning the preparation stages, again, a dip-coating methodology was followed utilizing a disc elevator for the layers' deposition. Firstly, surface modification of the used substrate (quartz, glass plate, Si wafers) was attained by its two-minutes double-immersion in the bacterial CP suspension. The followed LbL film preparation was attained by several cycles of CP-modified materials' immersions into: (i) $TiO_2$ suspensions and withdrawal after 2 min, and (ii) CP suspension again for 10 s. After the multiple immersions, the modified films were finally dipped into a HPW solution as to accomplish the photocatalyst's manufacture. FTIR and XPS results indicated that HPW is connected through the oxygen atoms to the $TiO_2$ surface and the binding between $TiO_2$ and CP is attained by Ti–O–P linkage, respectively. Moreover, SEM images illustrated the porous structure of the prepared films and the fine dispersion of the inorganic nanoparticles into or onto the polymeric surface (Figure 4). The final multilayered films

exhibited adequate photoactivity toward stearic acid (SA), crystal violet (CV) and MB under UV exposure.

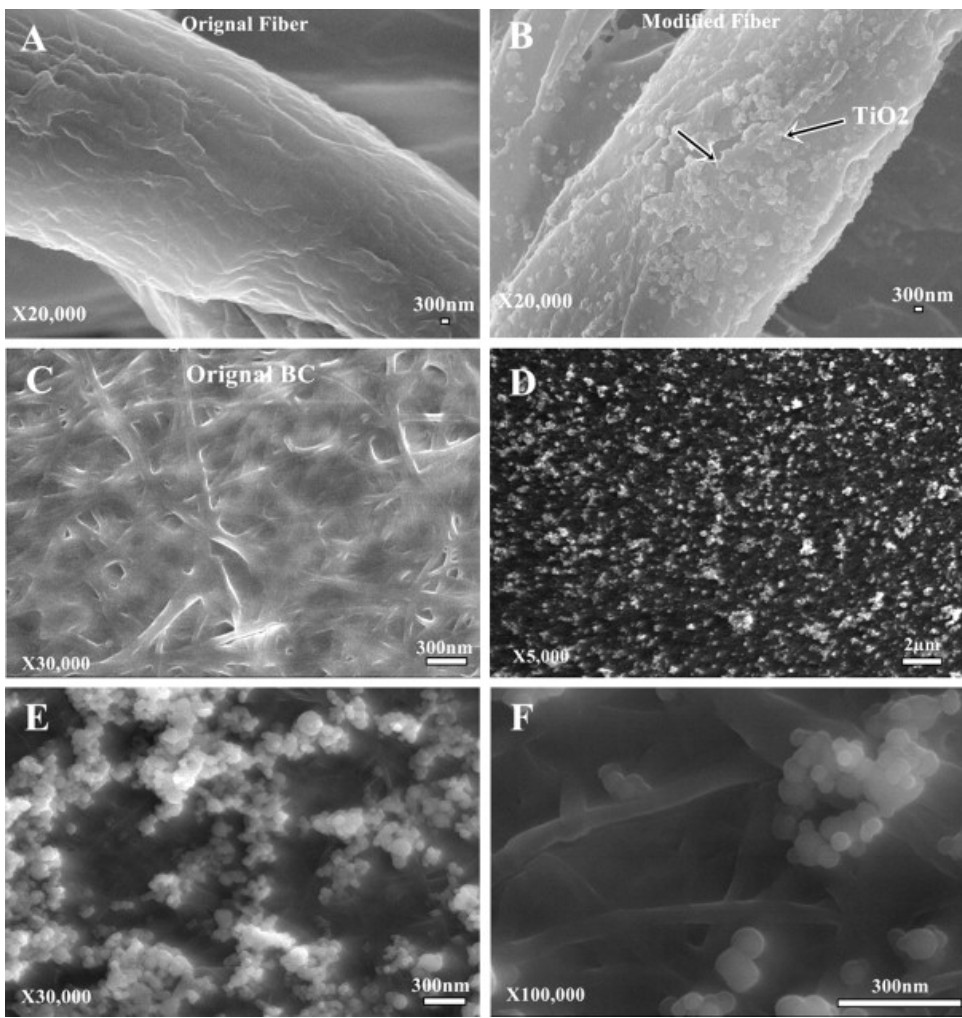

**Figure 4.** SEM images of unmodified fibers of cellulose filter paper (**A**) and after formation of CP/TiO$_2$ film (**B**), FEG/SEM image of original BC membrane (**C**) and after formation of CP/TiO$_2$ film on BC in low (**D**) and high magnification (**E**,**F**). Reprinted with permission of [30].

With another aspect of photocatalytic film fabrication, Farshchi et al. [4] prepared a biodegradable film comprised of the water-soluble carboxymethyl cellulose (CMC) modified with TiO$_2$–Ag nanoparticles and gelatin, via a film casting method. In brief, CMC and gelatin, in several concentrations, were dissolved in water at 70 °C and, after a mild cooling, glycerol as plasticizer agent was also added. An aqueous TiO$_2$–Ag latter dispersion was poured into the biopolymers' solution and magnetically stirred. The final solution was distributed into a polystyrene disc, dried and kept in a place without light for its further use. Results indicated that the TiO$_2$–Ag concentration did not affect to a great extent the thickness of the prepared films, whereas with the increasing content of gelatin and the inorganic nanoparticles, the films' thickness and moisture raised steeply, implying that the presence of the nanoparticles provoked the swelling of the gelatin after their interaction. Mechanical test measurements showed that with the increase of TiO$_2$–Ag content, the tensile strength of the modified films was also increased, exhibiting the strengthening effect of these nanoparticles to the polymeric matrix. Furthermore, the addition of gelatin seemed to enhance the elasticity of the modified polymeric films, mainly attributed to the moisture content of them. These novel and biodegradable films were tried for their photocatalytic activity against NH$_3$, ethanol and benzene.

From a different point of view, Wittmar et al. [31] employed a dropping cum separation method in order to prepare magnetic and porous cellulose macrospheres with photocatalytic properties, for water purification. $TiO_2$ and/or $Fe_3O_4$ nanoparticles were dispersed into an ionic liquid/dimethylsulfoxide (IL/DMSO) mixture with the aid of sonication, whereas cellulose was further added at the former inorganic nanoparticle dispersion. The nanocomposite particles were formed via a dropping separation method in a water bath, whereas the formed spheres were left in the means overnight, washed and finally freeze-dried. When the content of $TiO_2$ in the casting solution increased, a rise of both specific surface area and pore volume was also noticed for the prepared photocatalysts, which was proportional with the dilution of the ionic liquids with DMSO. Moreover, the thermal stability of the nanocomposite spheres doped with $TiO_2$ decreased in relation to the pure cellulose; a fact attributed to the relaxation of the intra- and intermolecular hydrogen bonds of the cellulose after the $TiO_2$ nanoparticles inclusion. The novelty of this study was the feasibility of the formed macrospheres (size 1–2 mm) concerning their recovery and reusability owned to the magnetic $Fe_3O_4$ nanoparticles, while the latter had no adverse effects on the photocatalytic efficiency of the composites.

A simple one-pot hydrothermal strategy was applied by Li and coworkers [10] who prepared a cellulose/$TiO_2$ composite film and aerogel for the removal of Cr(VI) and rhodamine-B. For the preparation part of cellulose nanofibril-derived carbon (CDC) integrated with $TiO_2$ (CDC/$TiO_2$), an amount of TEMPO-oxidized cellulose nanofibrils was dispersed in water with continuous stirring and ultrasounds. Afterwards, titanium isopropoxide was added into the former suspension and further kept for stirring. The obtained mixture was placed into an autoclave and heated (200 °C) for 2 days. For the preparation of the composite films and aerogels, an amount of bacterial cellulose and of the previously formed powders were dispersed in distilled water and rapidly stirred in order to obtain a homogeneous mixture. The use of this type of cellulose, herein, made the recovery of the produced material easy and feasible. For the films' preparation, a casting and freeze-drying step was also applied. According to the XRD results, the $TiO_2$ formed by this hydrothermal approach was in anatase phase while the prepared CDC/$TiO_2$ composite materials were polycrystalline.

With the aim to synthesize adsorptive and recyclable photocatalysts based on cellulose, a research team investigated a hydrolysis-precipitation method [32]. According to this study, TEMPO-oxidized cellulose nanofibers (TOCN) were firstly prepared using an oxidation method, while for the preparation of TOCN/$TiO_2$ composites, an amount of $TiCl_4$ was added in a TOCN suspension with adjusted pH at 2–3, under strong stirring. After the 30 min-stirring, the reaction temperature was raised at 80 °C, and the final pH was set at 7, with ammonia. The obtained precipitate was washed with water and ethanol, and finally dried, grounded and calcined for 2 h (550 °C, $N_2$). For the Cu-doped products, several amounts of $Cu(NO_3)_2 \cdot 3H_2O$ were added in the TOCN suspensions and then the previous process was followed, as well. XRD patterns showed that the addition of the TOCN and Cu did not provoke any change in the crystallinity of $TiO_2$, whereas TEM images showed the almost spherical shape and the uniform dispersion of the $TiO_2$ particles into the used matrix. $N_2$ adsorption–desorption isotherms indicated that the TOCN combination for the formation of mesoporous $TiO_2$ composite materials was beneficial for the enlargement of total pore volume and specific surface area of the polymeric nanofibers.

**Table 2.** Summary of the cellulose-supported $TiO_2$ photocatalysts enclosed in the presented literature.

| | Biodegradable Polymeric Matrix | | | | | Photocatalysis Parameters | | | Ref. |
|---|---|---|---|---|---|---|---|---|---|
| No. | Polymer Substrate | TiO$_2$ Precursor | Dopant | Immobilization Technique | Morphology of the Photocatalyst | Type of (Target) pollutant | Light Source | Degradation Efficiency (Time Required) | |
| 1 | Cellulose nanofibrils derived carbon + BC | Titanium isopropoxide (Ti(O-iPr)$_4$, 99%) | - | One-pot hydrothermal method | Composite film and aerogel | Cr(VI) Rhodamine B (RhB) | Xenon lamp 300 W (CEL-HXF300), λ = 420 nm, visible light | 100% (60 min) 100% (60 min) | [10] |
| 2 | Cellulose | TiO$_2$ (anatase) | Au nanoparticles | - Tape casting for the cellulose membrane - Simple suction filtration for the immobilization of Au and TiO$_2$ NPs | Composite membranes | Rhodamine B (RhB) | 500 W Xe light simulation of sunlight | ~90% (180 min) | [26] |
| 3 | Cellulose | TiCl$_4$ | Cu | Hydrolysis-precipitation method | Nanofibers | Organic dyes (reactive brilliant red K-2BP and cationic red X-GRL) | UV light, high pressure mercury lamp 300 w, λ = 365 nm | 96.57% (120 min) 99.73% (120 min) | [32] |
| 4 | Cellulose, cellulose triacetate | TiO$_2$ nanoparticles | GO | Phase inversion via immersion precipitation method | Membranes | Benzene toluene, ethylbenzene xylenes (BTEX) | UVC light–UV lamp 100 W, λ = 280 nm | ~80% (180 min) ~70% (180 min) ~90% (180 min) ~75% (180 min) | [28] |
| 5 | Cellulose | Titanium n-butoxide solution | Ag$_2$O nanoparticles | - Titania/cellulose composite by a surface sol–gel method - precipitation method for the nanocomposites | Hierarchical nanocomposites | Methylene Blue Rhodamine B Norfloxacin | UV light lamp 300 W | >90% (10 min) >90% (10 min) ~75% (10 min) | [29] |
| 6 | CMC | Commercial TiO$_2$-Ag nanopowder | Ag nanoparticles | Film casting of CMC with gelatin, glycerol | Composite film | NH$_3$ Ethanol Benzene | A closed system emmits light with λ = 254, 365, and 500 nm light | - - - | [4] |
| 7 | Cellulose | TiO$_2$ Aeroxide P90 nanoparticles | Fe$_3$O$_4$ nanoparticles | Dropping cum phase separation for the nanocomposite spheres | Magnetic cellulose macrospheres | Cu$^{2+}$ ions Rhodamine B | 2 × 15 W generates UV light λ = 365 nm | - 33-23% (60 min) | [31] |
| 8 | CAM | TiO$_2$ P25 paste | - | Dip coating method | 2D and 3D TiO$_2$ thin polymeric coated films | Cyanotoxins | Solar radiation simulator with a 1700 W air-cooled xenon arc lamp | 90% (12 kJ L$^{-1}$) | [9] |
| 9 | Cellulose phosphate | TiO$_2$ | - | Layer-by-layer (LbL) assembly technique | Porous multilayer films | Stearic acid Crystal violet Methylene Blue | UV radiation (16S Solar Light Simulator with a 150 W lamp | 95% (12 min) 95% (30 min) 50% (30 min) | [30] |
| 10 | Regenerated cellulose from recycled newspaper | Titanium-n-butoxide Ti(OBu)$_4$ | - | Phase inversion method | Nano-composite membrane | Phenol degradation | 30 W UV lamp (λ = 312 nm), visible light from a 30 W-LED λ > 420 nm | 96% in UV light (360 min), 78.8% in visible light (360 min) | [27] |

Abbreviations: CA, cellulose acetate; BC, bacterial cellulose; CMC, carboxymethyl cellulose; CAM, cellulose acetate monoliths; CDs, carbon dots; SWCNTs, single walled carbon nanotubes; GO, graphene oxide.

Photocatalytic Performance

Getting an insight into bibliography about the use of cellulose destined for the fabrication of photocatalytic membranes, it was shown that various pollutants were targeted for the examination of their possible removal. The majority of studies investigated the removal of dyes such as MB, RhB, MO, crystal violet, reactive brilliant red K-2BR and cationic red X-GRL. Their degradation was more than 90% in most of the cases and the experiments were carried out under UV light, sunlight, and visible light while the irradiation time ranged from 10 to 180 min [4,9,10,26–32]. Furthermore, other organic pollutants were explored with worth noticing results. Phenol, benzene, toluene, ethylbenzene and xylene were studied under UV and visible light irradiation, while their degradation efficiency ranged between 70% and 90% [28], [27] after 180–360 min of treatment. Moreover, other toxic pollutants were examined such as stearic acid, Cr(VI), and cyanotoxins microcystin-LR or cylindrospermopsin with remarkable photocatalytic results [9,10,30]. Concerning the photocatalytic substrates, various matrices of cellulose material were tested such as membranes, films, fibers, and magnetic macrospheres. Regarding the reusability of cellulose material in photocatalysis, the synthesized composites were tested for three to six repetition cycles, and as a result the photocatalytic materials showed exhibiting good stability.

Maryam Taghizadeh et al. investigated the photocatalytic activity of two types of membranes, cellulose triacetate with $TiO_2$ (CT), and cellulose triacetate with $TiO_2$ and GO (CTG). The results revealed that the photocatalytic activity of GTC was superior under UV illumination since the synergetic effect of $TiO_2$ and GO reduced the band gap energy of semiconductors [28]. In another notable study, different structures of photocatalytic materials were used including 2D P25 thin-films cellulose acetate monoliths (P25-CAM), $TiO_2$ thin-films cellulose acetate monoliths ($TiO_2$-CAM), and PET-monoliths $TiO_2$ loaded exterior paints (PETM). The PETM films had lower photocatalytic performance due to their limitation on mass transfer into their porous system and to the low adsorption of photons by the structure of material. Moreover, $TiO_2$-CAM films performed higher efficiency in comparison with P25-CAM due to higher amount of $TiO_2$ on their surface [9].

Zeo Lin et al. explored the photocatalytic activity of three novel $Ag_2O$-nanoparticle /$TiO_2$ nanocomposite materials supported by cellulose with 19.7 wt%, 33.9 wt%, and 51.8% wt content of $Ag_2O$, respectively. Researchers using these composites observed that the material with the lowest content of $Ag_2O$ presented the minimum band gap energy owing to the better formation of its structure, leading to the best photocatalytic performance in comparison with other materials. However, all materials showed an excellent photocatalytic performance in comparison with other cellulose manufactured composites and this performance can be attributed to their structure that was fabricated from the original stable cellulose substance, while the hierarchical construction of porous network with active combination of catalysts also contributed to the same result [29].

Regarding magnetic porous macrospheres manufactured from $TiO_2$, and/or $Fe_3O_4$, in the IL/DMSO mixtures, a notable study examined their photocatalytic activity. Accordingly, the macrospheres prepared with the lower quantity of DMSO exhibited an enhancement of the photocatalytic activity with a rise of the $TiO_2$ content. On the other hand, in the structures with higher amount of DMSO, the rise of catalyst content did not affect their performance due to the low distribution of the $TiO_2$ nanoparticles during the solvent casting method applied in the preparation stage. Regarding the latter, a more effective method should be applied in the future studies for a more efficient dispersion of the catalyst leading to an enhancement of the photocatalytic activity of material [31].

### 2.1.3. Alginate (Alg)
Synthetic and Characterization Routes

Alginate is a natural and anionic polysaccharide, mainly derived from the cell walls of algae. In recent years, the low price and abundance in nature, have rendered the alginate as a polymer of much attention. Due to the presence of hydroxyl and carboxyl groups on its molecule, alginate has been as well explored for the effective removal of metal ions. In fact,

its composition and sequence of 1,4-linked β-D-mannuronic acid (M) and 1,4 α-L-guluronic acid (G) units are considered to be the reasons of its adsorptive properties especially with divalent cations, in which it participates in intermolecular interactions. The monovalent salts of alginate are water-soluble. Sodium alginate is soluble in alkaline environment, and its aqueous solutions are stable in a pH range between 5 to 11. Nevertheless, with the addition of polyvalent cations, cross-linking reactions take place, and finally a thermo-irreversible gel is formed. Alginate is utilized in its salt form, mainly with calcium and sodium (depicted at Figure 5) and has been widely explored as a biodegradable polymer to fabricate photocatalytic materials with $TiO_2$ immobilized nanoparticles. A brief description of the studies reported herein for alginate-supported photocatalysts is presented in Table 3.

**Figure 5.** Possible binding mechanism of sodium or calcium alginate with $TiO_2$.

Nouri et al. [16] capitalized upon the manufacture of eco-friendly photo-biocomposite beads, utilizing calcium alginate and employing a green extrusion method. The prepared hydrogel composites were stored overnight in a $CaCl_2$ solution for the toughening of the beads and the day after the final gel beads were washed to remove the unattached $Ca^{2+}$ and further dried. From XRD it was showed that a small relocation of crystalline peaks to higher degrees confirmed the immobilization of the $TiO_2$, whereas SEM images illustrated the spherical morphology and the homogeneous distribution of the alginate beads. However, aggregation of $TiO_2$ particles was observed at the surface of the beads, while the less quantity of these inorganic particles in the final product possibly due to the stirring processes was also noticed. The rough surface of the composite hydrogel beads provoked after the photocatalytic process at the alginate-based material proved as a benefit for the adsorption of the basic blue 41.

Gelation techniques are commonly chosen for the preparation of alginate-based photocatalysts. Within an innovative effort, Dalponte et al. [33] reported on the synthesis of a buoyant alginate photocatalyst with immobilized $TiO_2$ nanoparticles for accomplishing both superior photocatalytic behavior and more feasible separation after treatment processes. The ionotropic gelation technique was employed, with $CaCO_3$ or $NaHCO_3$ gas-forming agents added firstly to a $TiO_2$ dispersion with the addition of a sodium alginate solution (Figure 6). After the stirring and sonication of the former mixture, the prepared solution was dropped into an acidic $CaCl_2$ solution with the aid of a pump. A crucial piece of the experimental part is located at the addition of a gas-forming agent to the alginate matrix, which aimed at the composite beads' density reduction and the production of floating photocatalysts. SEM/EDX, TGA, XRD, and FTIR results exhibited the uniform nature of the formed microbeads, possessing a rough surface and a porous morphology.

**Table 3.** Summary of the alginate-supported $TiO_2$ photocatalysts enclosed in the presented literature.

| No. | Biodegradable Polymeric Matrix | | | | | Photocatalysis Parameters | | | Ref. |
|---|---|---|---|---|---|---|---|---|---|
| | Polymer Substrate | $TiO_2$ Precursor | Dopant | Immobilization Technique | Morphology of the Photocatalyst | Type of (Target) Pollutant | Light source | Degradation Efficiency (Time Required) | |
| 1 | Calcium alginate | $TiO_2$ (P25, 20% rutile and 80% anatase, $\geq$99%) | GO | One-step emulsion gelation method | Hierarchical solid-liquid gel spheres | 2-naphthol Rhodamine B | A xenon lamp with power 500 W, and light intensity 100 mW/cm$^2$ | 50% (150 min) 58% (150 min) | [33] |
| 2 | SA | $TiO_2$ nanopowders, anatase phase | GO | Simple hydrothermal treatment method | 3D sodium alginate/graphene oxide/$TiO_2$ aerogel | Ibuprofen (IBUP) Sulfamethoxazole (SMX) | UV-A light through a photo-reactor with light 13.5 $\pm$ 1 W/m$^2$ | ~78% (90 min) ~90% (45 min) | [34] |
| 3 | CaAlg | Titanium (IV) oxide, anatase powder (99.8% trace metals basis) | - | Extrusion method | Biocomposite beads | Basic blue 41 | Cylindrical jacketed batch reactor with simulated sunlight irradiation 90 W/cm$^2$ | ~96% (240 min) | [16] |
| 4 | CaAlg | $TiO_2$ nanopowder P-25 | - | -Dripping method -Ionic gelation | Buoyant $TiO_2$/CaAlg photocatalyst | Tartrazine | UV 125 W), $\lambda$ = 254 nm | >89% (180 min) | [35] |
| 5 | SA | P25 (nanoscale $TiO_2$ powder) | - | - Homogeneous dispersion solution - Cross-linking | $TiO_2$-alginate composite aerogels as | Methyl orange | Simulated sunlight irradiation, 300 W xenon lamp | >96.7% (150 min) | [36] |
| 6 | SA | Titanium tert-butoxide | - | Precipitation method | Paper sheets modified with $TiO_2$/Sodium alginate nanocomposites | COD | UV light lamp with $\lambda$ = 246 nm | 42–18% (120 min) | [37] |
| 7 | CaAlg | AEROXIDE® $TiO_2$ P25 | - | Dry/wet spinning process | CaAlg/$TiO_2$ fibers | Methyl orange | Four UV lamps, PL-S/PL-L, 9 W), with $\lambda$ = 315–380 nm | ~98–90% (340–272 min) | [38] |

Abbreviations: SA, sodium alginate; CaAlg, calcium alginate; COD, chemical oxygen demand.

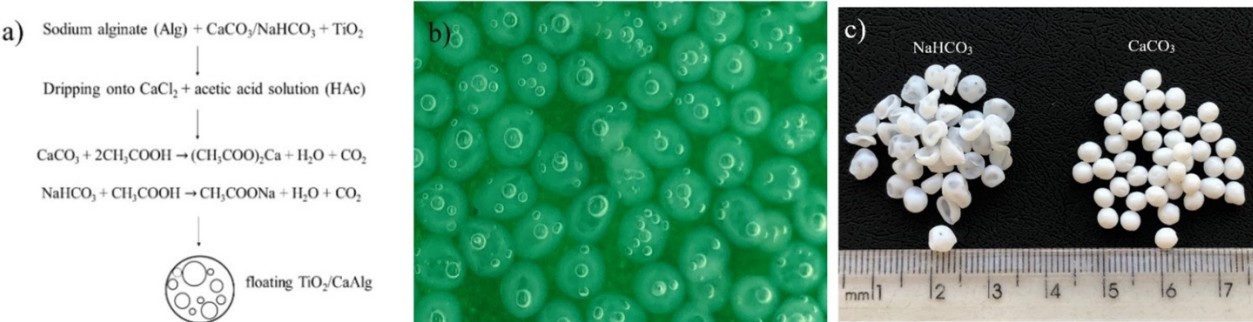

**Figure 6.** (**a**) Scheme from ionotropic gelation and $CO_2$ generation from both gas forming agents ($CaCO_3$ and $NaHCO_3$), (**b**) bubbles held within the gel beads with $CaCO_3$ in its formulation, and (**c**) floating $TiO_2$/CaAlg beads with 1:1 ratio of $NaHCO_3$/alginate and 1:1 ratio of $CaCO_3$/alginate [33].

In an attempt to fabricate calcium alginate gel spheres for the immobilization of $TiO_2$ nanoparticles, Lan et al. [34] employed a one-step emulsion gelation method with graphene oxide (GO) as a dopant for the stabilization of the formed photocatalyst. A brief illustration of the experimental part is depicted at Figure 7. The use of ionic liquid into the emulsion drops would enable the extraction and removal of the organic pollutants, whereas the prepared elastic gel spheres were considered to be feasible in the recycling processes after treatment. In this case, the calcium alginate gel acted as a flexible matrix to modify and protect the $TiO_2$ nanoparticles from a possible demulsification process.

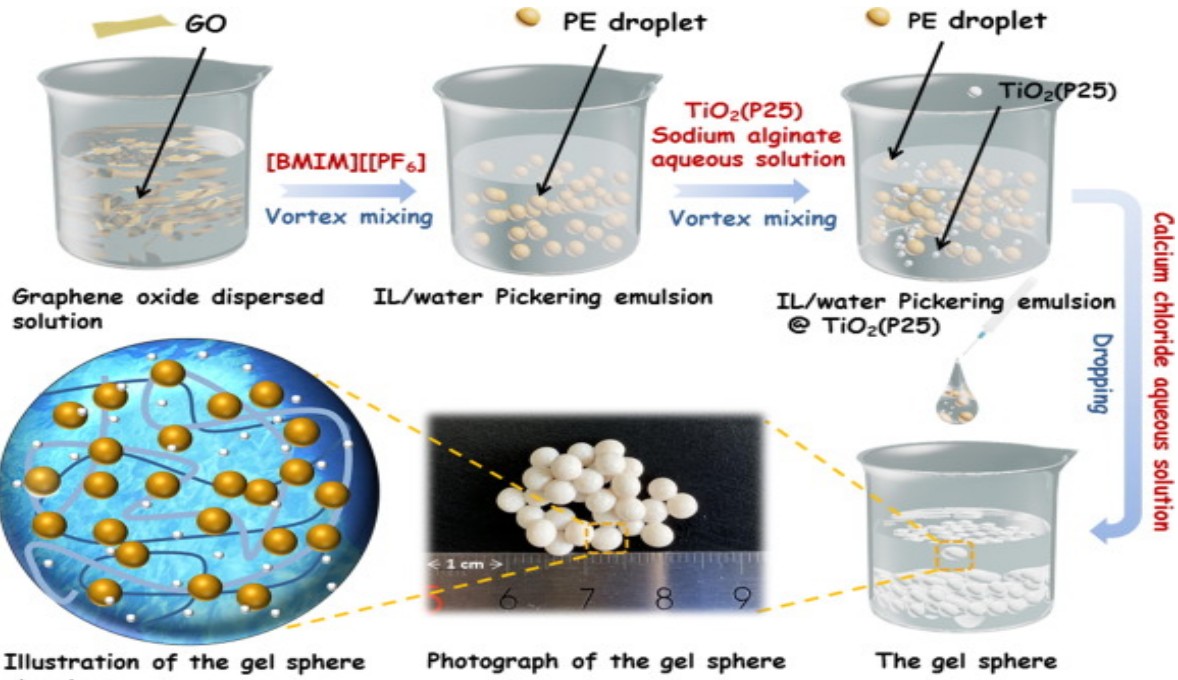

**Figure 7.** Schematic of the preparation process of the gel sphere [34].

The preparation of aerogels with immobilized $TiO_2$ nanoparticles for wastewater remediation practices is an additional application of alginate salts. Exploring once again homogeneous dispersion and ionic crosslinking pathways, Dai et al. [35] fabricated $TiO_2$-alginate aerogels for oil/water separation. After the formation of hybrid sodium alginate-$TiO_2$ sol with the aid of ultra-sonication, the sol was freeze-dried, while the formed hybrid

aerogel was immersed latter into a CaCl$_2$ solution for the cross-linking step. An additional freeze-drying stage was followed after the washing of cross-linked aerogel. XRD patterns showed a similarity of the composite aerogels' diffraction peaks to those of the pure TiO$_2$, implying that all the followed treatment steps did not affect the crystal form of TiO$_2$ nanoparticles. SEM images (Figure 8) also showed that the formed aerogels exhibited a 3D porous design, while authors proposed that the rough surface of the sodium alginate-TiO$_2$ aerogels accompanied with the high affinity to water would be beneficial for its purpose's construction.

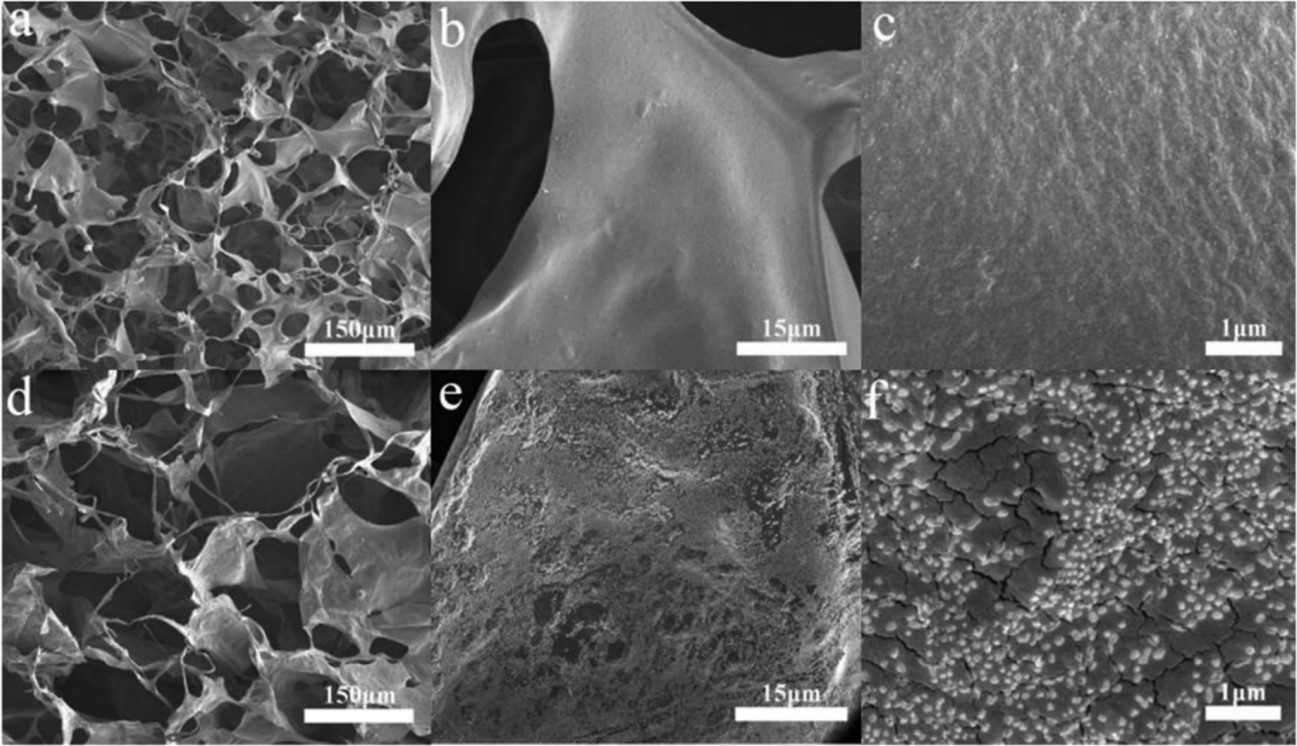

**Figure 8.** SEM images of the as-prepared SA aerogel (**a–c**) and SAT aerogel (**d–f**) with different magnifications. Reprinted from Reference [35].

Within a similar conceptualization, Nawaz et al. [36] prepared 3D-structured sodium alginate (SA) aerogels with the incorporation of reduced graphene oxide, aiming at the better bonding between the TiO$_2$ and sodium alginate. The synthesis of the aerogels was based on a hydrothermal treatment method. Here, sodium alginate played a crucial role in the stability of the formed aerogels, since the electrostatic forces between GO and the SA, enabled the well dispersion of the GO into the polymeric solution. The three-dimensional composed aerogels were considered as floating and recyclable photocatalytic materials since the degradation efficiency of the composite materials was not severely affected after five photocatalytic cycles.

Within a green pathway, Abdel Rehim et al. [37] prepared modified paper composites after blending paper with sodium alginate/TiO$_2$ nanocomposites in several ratios. Sodium alginate possessed a role of template and directed the design of the formed composites. The synthetical route for the sodium alginate/TiO$_2$ composite was based on a precipitation method, whereas the paper composites were prepared with the aid of a hydraulic press. SEM images revealed that as the content of TiO$_2$ increased, the aggregation of the nanoparticles was also increased. It was confirmed that the presence of sodium alginate as biopolymer boosted adhesion of nanoparticles to paper sheets and thus limited the harmful impact of the photocatalyst on them.

Dry/wet spinning method is another process utilized for the preparation of porous alginate porous fibers with dispersed and finally immobilized TiO$_2$ nanoparticles as to design pre- or post-treatment photocatalysts destined for membrane filtration technology. Specifically, this method is based on the drying of hydrogel fibers (supercritical CO$_2$ conditions, 45 °C, 100 bar), synthesized via a spinning protocol. In the case of Papageorgiou et al. [38], the use of calcium alginate/TiO$_2$ hybrid fibers as a preliminary step to the membrane treatment for methyl orange removal, led to a triple improvement of the removal's efficiency.

Photocatalytic Performance

Getting an insight into bibliography of alginate-based materials with combination of TiO$_2$ nanoparticles, it is observed that they were used for various applications, such as the photo-degradation of common dyes (MO, basic blue 41, tartrazine, RhB), removal of pharmaceuticals (ibuprofen, and sulfamethoxazole), and other organic compounds (2-naphthol). Moreover, alginate-TiO$_2$ nanocomposites were used in different morphologies including fibers, membranes, hydrogel spheres, and papers. The irradiation was carried out under two main sources, UV light or sunlight for a time range between 45–340 min. Photocatalytic materials based on alginate showed a high recyclability with a range of 3–7 cycles [16,33–38]. Furthermore, for the better understanding of the exporting photocatalytic results and merits of alginate matrix, selected studies are presented below.

According to Papageorgiou et al., CaAlg-TiO$_2$ fibers were manufactured, and their photocatalytic activity was examined on common azo dyes. The alginate fibers proved to be much more efficient than the examined bulk TiO$_2$ powder, owing to the high distribution and stability of TiO$_2$ nanoparticles into the polymeric structure, which presents a high surface area thus leading to an enhanced photocatalytic activity. Moreover, the stability of the photocatalytic material was also checked with promising results, as the structure of the material was not affected by UV irradiation after numerous hours of illumination, [38]. Lan et al. investigated the degradation efficiency of hydrogel spheres which were fabricated by the combination of sodium alginate as polymeric material and GO/TiO$_2$. Considering the observed results, it was concluded that gel spheres had a large adsorption capacity and a high photodegradation efficiency since they can work as a microreactor in static phase, while TiO$_2$/GO nanoparticles can offer a high surface area contributing to the same result. Herein, the matrix was favored in the ionic liquid phase due to the high adsorption of ionic liquids onto its structure, and its high surface area worked perfectly on oil/water interface conditions improving the photocatalytic activity. Moreover, after applying a continuous flow system, researchers achieved high degradation results in comparison with existing literature since the hydrogel spheres operated like column filler, creating a continuous photocatalytic system [34].

Furthermore, reduced graphene oxide-TiO$_2$/SA (RGOT/SA) aerogels were applied for the degradation of pharmaceuticals with astonishing results. Mohsin Nawaz et al. fabricated this 3D structured material which performed an enhanced photocatalytic activity. More specifically, the large specific area and the efficiency to adsorb a significant amount of UV illumination contributed to an enhanced degradation of the target pollutants. Additionally, according to reusability of RGOT/SA aerogel, five cycles of experiments were carried out exhibiting high stability due to the interconnected context of the 3D structure which blocks the heaping of fractional units. RGOT/SA material showed also great elasticity and deformation ability, while the gel material without SA was fragile and easily broken with an increase of pressure [36]. Juguo Dai et al. examined the degradation of MO with sodium alginate/TiO$_2$ (SAT) aerogel. The observed high degradation efficiency was attributed to the excellent distribution of TiO$_2$ onto SAT aerogel structure. Moreover, the alginate substrate covered with photocatalyst was characterized as durable against photoinduced corrosion and it could be reused for six photocatalytic cycles [35].

### 2.1.4. Starch

Synthetic and Characterization Routes

Starch is a renewable material with biocompatible and biodegradable nature, low cost and high abundance, and due to this positive profile, it has been widely explored in food, textile, packaging as well as pharmaceutical industries. It mainly consists of amylose and amylopectin, while one of the most challenging chapters concerning its use is its dissolution since the strong inter- and intra-molecular hydrogen bonds and its semicrystalline nature with double helices, require strong polar systems. Biodegradation of starch can mainly proceed via hydrolysis at the acetal bonds by enzymes.

Concerning the utilization of starch for the fabrication of polymer supported-$TiO_2$ materials solely, the literature data is more than scarce. A study found in the bibliography utilizing starch in this way was reported by Lin et al. [39], thus using a mixture of starch with poly(vinyl alcohol) (PVA) with the aim to fabricate starch/PVA/nano-titania (S/P/T) composite photocatalytic membranes. The methodology followed for the preparation of the eco-friendly composite films was based on flow casting and vacuum drying methods. The authors implied that the stability, whiteness, water solubility, barrier properties, photocatalytic performance, as well as the antibacterial profile of the prepared membranes was greatly enhanced after the incorporation of the nano-titania. The latter fact may be attributed to the covering of the space between the starch and PVA molecules in the substrate by the inorganic nanoparticles, while the migration of oxygen and moisture was also enabled, and thus, the physical and mechanical properties of the final membrane was enhanced.

With a different approach, Muniandy et al. [40] utilized starch as a template to form mesoporous anatase $TiO_2$ nanoparticles via a green synthetic pathway. A sol-gel method was applied by, briefly, dispersing an amount of titanium isopropoxide into an aqueous starch solution (stirring, 85 °C) and precipitating the $TiO_2$ nanoparticles after the slow addition of an ammonium hydroxide solution. Results showed that the starch was removed from the final particles after calcination, while its role as a crucial agent in the formation of mesoporous nano-titania was also implied. Regarding the fabrication mechanism of the mesoporous $TiO_2$, the initiation of crystal growth was located at the diffusion of the $Ti^{4+}$ which formed complexes with the amylose counterparts between the already swollen starch granules. The anatase $TiO_2$ started to self-assembly due to the Van der Vaals forces between the surficial molecules and finally formed uniform and compact structures over the starch microspheres. Another study reported on the use of starch as a natural means for preventing the agglomeration of nano-titania particles and modifying the morphology and the photocatalytic behavior was addressed by Bahar Khodadadi [8]. Herein, Nd–Ce-codoped $TiO_2$ nanoparticles were synthesized via a sol-gel route, whereas starch was initially used as a green means for the dispersion of the titania precursor.

However, two works were found in the literature utilizing starch as an abundant carbon doping agent for the preparation of C-doped $TiO_2$ nanoparticles, with enhanced photocatalytic activity. According to Warkhade et al. [41], a low temperature and eco-friendly sol-gel method was applied, facilitating starch as a doping as well as capping factor, whereas the C-doped photocatalysts exposed superior photocatalytic degradation of rhodamine B in contrast to the undoped pristine ones. Within a similar conceptualization, C-doped $TiO_2$ nanoparticles were fabricated using starch as a carbon source and a hot air process for manufacture pathway [42]. Briefly, an aqueous dispersion of $TiO_2$ was well-mixed with different concentrations of starch, then inserted into a hot air oven (180 °C, 4 h) and finally calcined. Again, the photocatalytic behavior of the C-doped product toward phenol degradation was enhanced in contrast to the undoped particles.

Photocatalytic Performance

Regarding the literature of starch-based $TiO_2$ materials for the removal of different pollutants from wastewater through photocatalytic process, it was noticed that researchers mainly focused on MB and rhodamine B (RhB) dyes removal, widely used in textile industry.

In most studies UV light and/or sunlight radiation was employed for the degradation of different initial concentration of dyes, ranging from 4 to 50 ppm [8,39–42]. The removal of contaminants ranged within 94–100% in most of the cases, and treated solutions were exposed to irradiation for 90 to 360 min.

Moreover, it was noted that the structure of starch/$TiO_2$ materials doped with various inorganic compounds led to an increase of their photocatalytic activity. In most of the studies, carbon-doped $TiO_2$ nanoparticles were examined for photocatalytic activity, while starch/PVA-$TiO_2$ was also used in one study. Concerning the forms of photocatalytic materials, films and membranes were mainly synthesized [39], while the best performance was achieved by the materials with the largest surface area [8,40,42].

Doping of $TiO_2$ also played a beneficial role on the photocatalytic activity of the synthesized materials. Khodadabi et al. examined $TiO_2$ nanoparticles coupled with Ce and Nd and starch forming a nanocomposite material, demonstrating results which show that the dopant ions can affect the $TiO_2$ band gap, enhancing thus the degradation efficiency of the tested pollutants. According to the authors, the material with 1% of dopant ions had the best performance but increasing the quantity of dopants ions led to a deterioration of the degradation efficiency as they can be placed on sites where $TiO_2$ nanoparticles are located, thus reducing the surface area of photocatalytic material, and preventing the adsorption of irradiation. [8]. In the same way, carbon doped-$TiO_2$ combined with starch, degraded phenol under irradiation by sunlight due to their small-sized crystalline content and the large specific area. The highest degradation efficiency was achieved by the composite with the minimum crystalline size (13.78 nm) among the five manufactured polymeric materials [42].

In another interesting study, researchers fabricated a novel material with superb photocatalytic activity using starch as template combined with $TiO_2$ nanoparticles, employing titanium tetraisopropoxide (TTIP) which played a crucial role in photocatalytic activity of the nanocomposite material. Particularly, the rise of TTIP concentration during the synthesis of material and the change of pH from basic to acid, increased the particle size of the nanocomposites, reducing their active surface area. Consequently, their photocatalytic activity decreased, too. Moreover, the synthesized composites were very stable during photolytic experiments, up to ten cycles, keeping a superior degradation efficiency of MB. The nanocomposite material was very stable even after ten cycles, while researchers used a washing step with boiling water prior to each reuse. The latter step may have contributed on the increase of their reusability [40].

### 2.2. Synthetic Polymers with Biodegradable Nature

### 2.2.1. Poly(Lactic Acid) (PLA)

Synthetic and Characterization Routes

PLA is one of the most dynamic and promising biodegradable synthetic polymers derived from renewable resources, such as corns, sugar beets, wheat and other starch-based products. It can be synthesized mainly by the polycondensation of lactic acid or by ring opening polymerization of lactide with the aid of a catalyst. PLA is totally degraded under compost conditions. Except its wide use in pharmaceutical technology, the fabrication of PLA for the synthesis of $TiO_2$-immobilized photocatalytic materials has limited literature, which is presented below, in brief. The synthetic pathways for these photocatalysts include mainly casting, electrospinning and spin-coating methods. The studies for PLA-supported photocatalysts included in the present review are presented in Table 4.

**Table 4.** Summary of the PLA-supported $TiO_2$ photocatalysts enclosed in the presented literature.

| | Biodegradable Polymeric Matrix. | | | | | Photocatalysis Parameters | | | |
|---|---|---|---|---|---|---|---|---|---|
| No. | Polymer Substrate | $TiO_2$ Precursor | Dopant | Immobilization Technique | Morphology of the Photocatalyst | Type of (Target) Pollutant | Light Source | Degradation Efficiency (Time Required) | Ref. |
| 1 | CA PCL PLA | Aeroxide® P25 $TiO_2$ nanoparticles | - | Solution casting method | Composite films | Methylene blue (MB) | UV-A light system fitted with four 40 W lamps | 72% (180 min) | [3] |
| 2 | CS + PLA | Titanium dioxide (Degussa P90) | - | - Sol-gel method, - step wise spin-coating method | Hybrid Multilayer Coated films | Methyl orange | UV light radiation under a UV instrument | ~80% (600 min) | [6] |
| 3 | PLA | $TiO_2$ nanoparticles (anatase) | - | Electrospinning | Membranes of hybrid nanofibers | Ampicillin | UV light through a lamp 120 W | 54–34% (30 min) | [17] |
| 4 | PLA | Tetrabutyl titanate (TBT, 98% pure) | - | Electrospinning | $TiO_2$/PLA composite nanofibers | Methylene orange | UV lamp, λ = 375 nm | ~40% (240 min) | [43] |
| 5 | PLLA | Titanium dioxide P25 (~80% anatase and 20% rutile) | - | Phase inversion method | GO/$TiO_2$ PLLA-supported nanocomposite films | Sulfamethoxazole Sulfadiazine Levofloxacin Norfloxacin Moxifloxacin Isoniazid Metronidazole Lincomycin Trimethoprim | Simulated solar irradiation through Suntest Atlas CPS+ solar with a xenon lamp (1.5 W and 750 W/m$^2$) | >90% for most of antibiotics (120–360 min) | [44] |

Abbreviations: CS, chitosan; PCL, polycaprolactone; PLA, poly(lactic acid); PLLA, poly(L-lactic acid).

The photocatalytic performance of three different biodegradable polymers, including poly(lactic acid) (PLA), polycaprolactone (PCL) and cellulose acetate (CA), against methylene blue was evaluated by Jing Xie and Yen-Con Hung [3]. According to this study, a simple solution casting method was employed for the synthesis of different biodegradable polymer-assisted structures, with TiO$_2$ incorporated nanoparticles. TiO$_2$ nanoparticles were dispersed into the proper solvents selected specifically for each polymer type, with the aid of ultra-sonication. A volume of the TiO$_2$ suspension was further added into the polymeric solution and the final dispersions were deposited into petri dishes and dried. SEM images (Figure 9) illustrated the uniform distribution of TiO$_2$ nanoparticles onto CA and PLA films, while this behavior was not noticed in the case of the PCL films in which a poor compatibility between inorganic particles and polymeric matrix was observed. Concerning the photodegradation rate, it was raised in the case of CA and PCL materials, with increasing the TiO$_2$ content in the prepared structures.

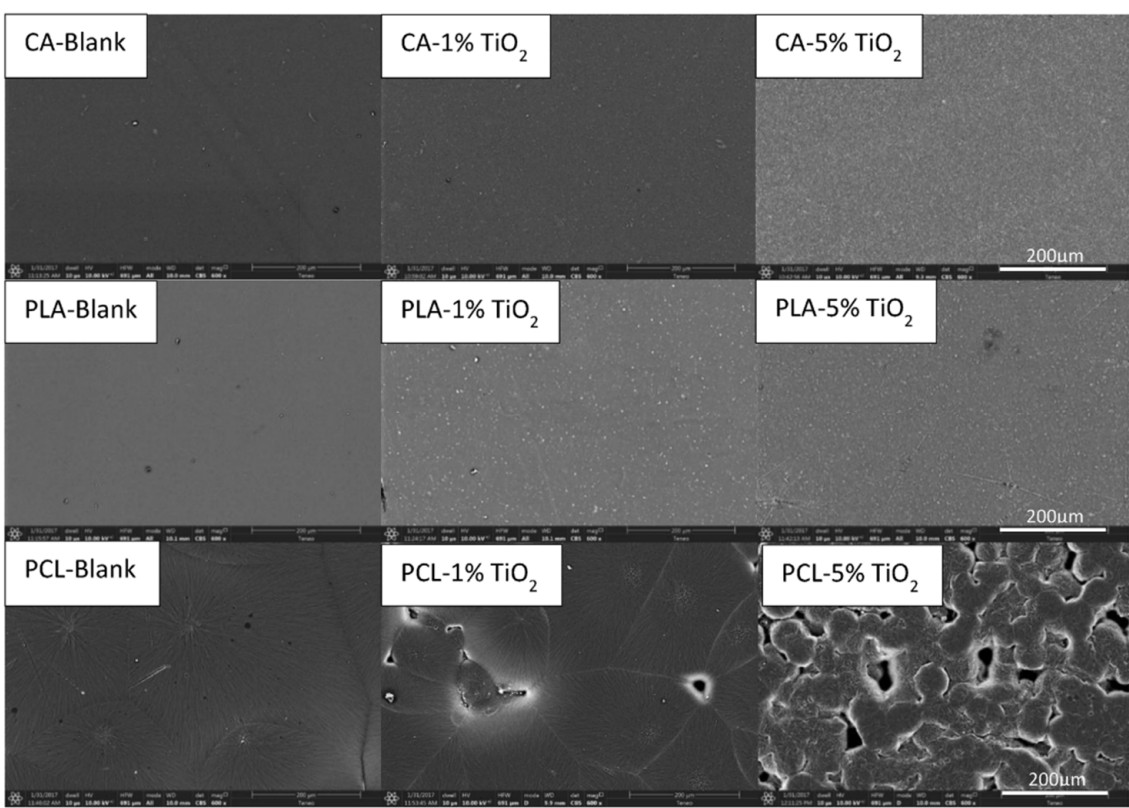

**Figure 9.** SEM images of TiO$_2$ NPs embedded PLA, PCL and CA polymeric films. Reprinted from Reference [3].

With a really interesting approach, Zhu et al. [6] manufactured a hybrid multilayer coating applied to a PLA film, by employing a spin-coating method. The novelty herein not only was located on the use of this methodology for the design of a multilayer photocatalyst (Figure 10), but also on the surface fluorination of the TiO$_2$ nanoparticles in order to ameliorate their dispersion. Between the PLA film and the upper inorganic photoactive layer consisted of the fluorinated TiO$_2$ nanoparticles, a middle layer of a hybrid material comprised by chitosan crosslinked with 3-glycidyloxypropyl trimethoxy silane was applied. The use of the latter aimed at the increase of the affinity between the organic PLA layer and the upper inorganic TiO$_2$ coating. For the preparation of the middle coating of photocatalytic materials, the hybrid chitosan sols were diluted in ethanol and applied on PLA thermomoulded membranes, by utilizing a spin coating equipment. Furthermore, the photocatalytic coating was applied on the surface of the former films with the aid of spin-coating again. The final transparent films had a thickness of approximately 4–6 μm, while their photocatalytic performance for the removal of MO was further investigated.

The results indicated that the degradation of PLA layer was pointedly restrained due to the UV adsorptive behavior of the multilayer coating.

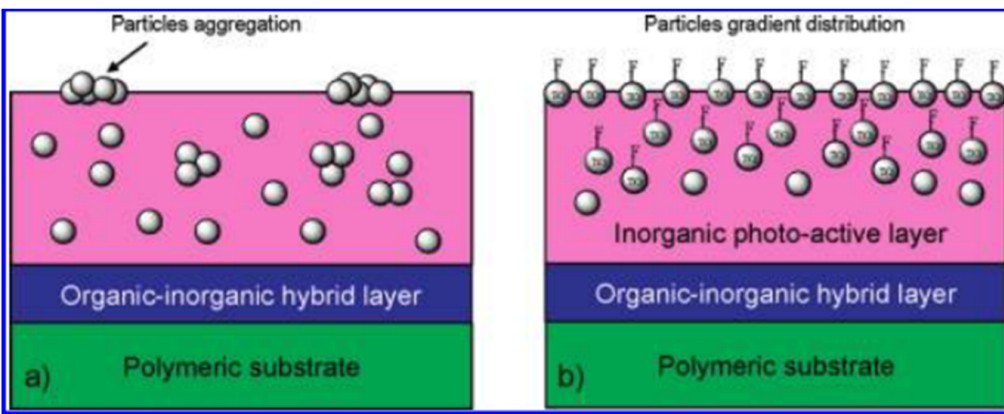

**Figure 10.** Schematic illustration of the multilayer functional coating on PLA substrate with (**a**) TiO$_2$ nanoparticles without surface fluorination, (**b**) surface fluorinated TiO$_2$ nanoparticles [6].

By employing the electrospinning method, PLA was also applied as a polymeric nanofiber matrix for the immobilization of TiO$_2$ particles. In a relative study, PLA/TiO$_2$ hybrid fibers were prepared on fiberglass substrates, while the photocatalytic efficiency of the obtained material was tried against ampicillin [17]. For the better adherence of the nanofibers onto the used supports, the surface of the latter was covered with PLA before utilizing the electrospinning process. With a similar point of view, Liu and coworkers [43] prepared also PLA/TiO$_2$ composite nanofibers for the degradation of methylene orange, via an electrospinning pathway. Herein, TiO$_2$ nanofibers (diameter between 600–700 nm) were manufactured after calcination of the as-spun PLA/TiO$_2$ composite nanofibers in air conditions (550 °C, 3h), while PLA had only a role of polymeric template.

With an interesting approach concerning the combination between different types of biodegradable polymers, Kreetachat et al. [7] investigated the design of bio-composite films destined for toluene removal. According to the authors, TiO$_2$ was added in different concentrations to a blend containing three different biodegradable polymers such as PLA, poly(butylene adipate-co-terephthalate) (PBAT) and poly(butylene succinate) (PBS). The inorganic/polymeric composite bio-films were prepared via a blown film method with a thin screw extruder (160 °C), while the obtained films had a thickness of 40 μm. SEM images showed the well dispersion of TiO$_2$ particles onto the polymeric matrix, but a possible aggregation of the crystalline particles during preparation stage was also implied.

Another novelty in the immobilization of TiO$_2$ nanoparticles in a biodegradable polymeric matrix was reported by Malesic Eleftheriadou et al. [44], whose study employed poly(L-lactic acid) (PLLA) as a supporting material for the fabrication of nanocomposite GO/TiO$_2$ films. Herein, the selection of GO was based on its beneficial nature in the adsorbance of several contaminants and as electron acceptor. This was the first attempt to explore the photocatalytic performance of PLA/TiO$_2$ based materials against emerging contaminants, such as antibiotics. The manufacture of the composite films was attained by a phase inversion method, according to which PLLA was firstly dissolved in a trifluoroacetic acid (TFA)/chloroform mixture, while the latter viscous solution was deposited into a glass casting plate as to prepare the desired film dimensions. For the GO and TiO$_2$ particles, which were dispersed under sonication, the same protocol was followed. According to the authors, the surface and porous characteristics of the prepared membranes stayed almost unaffected after the fourth photocatalytic cycle, implying the stable character of PLLA matrix after several runs.

Photocatalytic Performance

In the field of bio-based polymers, PLA has also been used for the fabrication of $TiO_2$ composite materials with photocatalytic activity. Three interesting studies used the synthesized composites for the degradation of pharmaceuticals (mainly antibiotics) [17,44,45], while another group of studies examined the degradation of common dyes such as MB and methylene orange [3,6,43]. The polymeric materials, acting as supports for the photocatalysts, were mainly manufactured in two morphologies: films and nanofibers. The degradation efficiency of pollutants ranged within different levels. The degradation of pharmaceuticals was near to 90% in most of cases, while the photocatalytic degradation of the azo dyes varied mostly between 70% and 90%. The photocatalytic experiments were carried out under UV and sunlight irradiation, while an adsorption step was priorly applied. A wide range of irradiation times was applied starting from 2 to 10 h. However, in most cases, the photocatalytic treatment lasted for 4 to 6 h. Finally, the recyclability of materials was evaluated ranging from 2 to 10 cycles.

More specifically, Malesic Eleftheriadou et al. studied the degradation of an antibiotics' blend under simulated sunlight radiation, using a PLLA nanocomposite film which was fabricated in combination with GO and $TiO_2$. Films were fabricated with different quantities of $TiO_2$. The best performance was achieved by the film with the larger quantity of catalyst, (PLLA-GO-50 wt% $TiO_2$) compared to the others which included lower amounts of $TiO_2$. This result can be explained from the production of hydroxyl radicals, as with increase of catalyst their production was increased leading to a higher degradation efficiency. Moreover, the films were examined in four cycles of photocatalytic experiments without significant differences observed in their photocatalytic efficiency or on their surface. Although a slight removal of $TiO_2$ was detected after the second run, the surface and its porosity remained unaffected from the additional runs [44]. In another notable study, the same PLLA-GO-$TiO_2$ nanocomposite film was used as in the previous study and a biobased PET-GO-$TiO_2$ was also examined for its photocatalytic performance. According to results, the biobased PET-GO-$TiO_2$ achieved a higher photocatalytic activity. Since in both composites the amount of $TiO_2$ incorporated was the same, the different catalytic performance was attributed to the characteristics of the polymers. PLA fabricated films are more skin-tight films compared to PET which forms more foamed structures, and as a result the active surface area of the former is smaller leading to a less efficient photocatalytic activity [45]. According to another study, researchers manufactured hybrid photocatalytic membranes using PLA/$TiO_2$ electrospun nanofibers supported on fiberglass substrate in order to examine their photocatalytic activity against antibiotics. The type and structure of the several fiberglass utilized for the construction of the hybrid polymeric networks greatly affected on the architecture of the final PLA/$TiO_2$ materials. It was observed that the fiberglass did not embed perfectly on photocatalytic membranes, while a superior efficiency of ampicillin removal was achieved with the fiberglass fabric plain woven-type photocatalyst. Thus, the fewer quantity of $TiO_2$ led to poorer photocatalytic performance. Moreover, the low degradation efficiency was caused by the breakage of PLA/$TiO_2$ nanofibers under the experimental conditions [17].

Furthermore, in another interesting approach Zhu et al. investigated the photocatalytic performance of PLA-$TiO_2$ composites coated with chitosan siloxane hybrids. Their photocatalytic activity was really high achieving a superb performance on the degradation of methyl orange dye (MO), while their recyclability was tested for 10 cycles. Among the three different fabricated films, the best results were extracted by the PLA films with the incorporated fluorinated $TiO_2$ particles due to the more effective distribution of catalyst. Moreover, the addition of fluorinated $TiO_2$ on the top of PLA films, limited the breakage of PLA under visible light and led to a high photocatalytic activity [6].

### 2.2.2. Polycaprolactone (PCL)

Synthetic and Characterization Routes

PCL (polycaprolactone) is a semi-crystalline aliphatic polyester produced by a ring-opening polymerization of ε-caprolactone, mainly in the presence of tin octanoate as catalyst. It is soluble in many organic solvents, while it possesses a semi-rigid nature in room temperature conditions. Enzymes and fungi simply degrade PCL, while for the enhancement of its biodegradation several copolymers with lactide or glycolide are proposed. Due to its low melting point, advanced rheological and viscoelastic characteristics, PCL has been extensively explored in the manufacture of electrospun porous fibers. Only recently PCL was reported as a template to prepare PCL/TiO$_2$ fibrous mats as dynamic candidates for efficient photocatalysts and thus further exploration in this field should be investigated. A brief description of the PCL-supported TiO$_2$ photocatalysts discussed herein is presented in Table 5.

In this context, Karagoz et al. [46] fabricated Ag and TiO$_2$ onto PCL electrospun nanofiber (NF) mats for the photocatalytic degradation of methylene blue, amongst their other perspective properties, such as the antibacterial and surface-enhanced Raman spectroscopy (SERS) application. According to SEM images (Figure 11), all the prepared mats exhibited uniform profile, while the composite PCL/TiO$_2$–Ag showed the larger fiber diameter. The latter fact may be attributed to the effect of TiO$_2$, and Ag nanoparticles accompanied by the increase of the viscosity in the solution with the most components that hindered the ion mobility throughout the applied electric field. With an aim to reduce the diameters of fibers and expand the specific surface area, Tu et al. [12] incorporated also rectorite (REC) to manufacture porous electrospun PCL fibrous mats, via electrospinning. REC, which is a type of layered silicate, has lately gained ground for the synthesis of TiO$_2$ nano-composites owing to their synergistic effects for advanced photocatalytic performance.

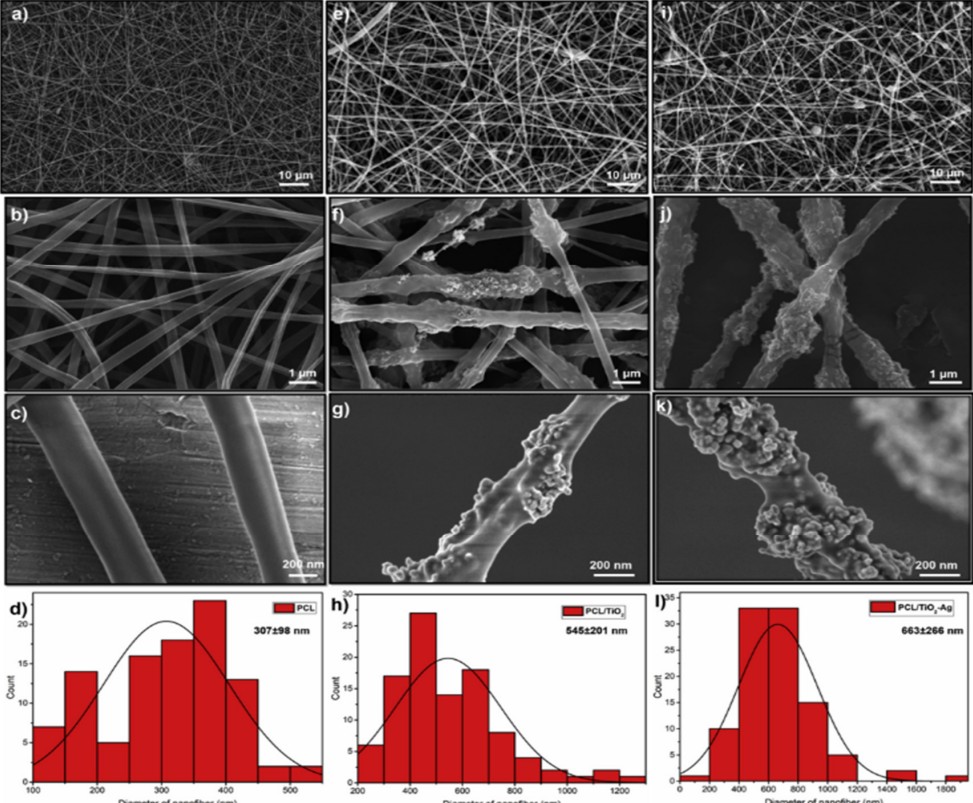

**Figure 11.** SEM images of (**a**–**c**) PCL, (**e**–**g**) PCL/TiO$_2$, (**i**–**k**) PCL/TiO$_2$-Ag NFs. Histogram presented the diameter distribution of (**d**) PCL, (**h**) PCL/TiO$_2$, (**l**) PCL/TiO$_2$-Ag NFs. Reprinted with permission of [46].

**Table 5.** Summary of the PCL-supported $TiO_2$ photocatalysts enclosed in the presented literature.

| | Biodegradable Polymeric Matrix | | | | | Photocatalysis Parameters | | | Ref. |
|---|---|---|---|---|---|---|---|---|---|
| No. | Polymer Substrate | $TiO_2$ Precursor | Dopant | Immobilization Technique | Morphology of the photocatalyst | Type of (Target) Pollutant | Light Source | Degradation Efficiency (Time Required) | |
| 1 | PCL | AEROXIDE® $TiO_2$ P25 nanopowder | Ag | One-step electrospinning | Ag and $TiO_2$ modified PCL electrospun nanofiber mats | Methylene blue Ibuprofen *E. Coli* *Staphylococcus aureus* | UV lamp with wavelength 380 nm and intensity 400 W | 93–90% (160 min) 50% (50 min) - | [46] |
| 2 | PCL | $TiO_2$ nanoparticles (P25) | REC | Electrospinning | PCL/$TiO_2$/REC porous mats | Rhodamine B | Exposure to a UV lamp with λ = 254 nm and intensity 25 W | 98% (240 min) | [12] |
| 3 | PCL | $TiO_2$ powder | - | Electrospinning & plasma treatment | PCL/$TiO_2$ nanofiber mats | Reactive Black 5 | UV lamps with λ = 254 nm | ~50% (120 min) | [11] |
| 4 | PCL | n-$TiO_2$ powder | - | Solvent-cast process | n-$TiO_2$ immobilized PCL films | Methylene blue, C. albicans | UVA lamp (365 nm) used or a visible light emitting sodium lamp | 94% (150 min) 54% (60 min) | [5] |
| 5 | PCL | n-$TiO_2$ powder | - | Solvent-cast process | n-$TiO_2$ immobilized PCL films | 4-chlorophenol | UV lamp (Kolorlu× H400/40 400 W 0303 Hungary, λ = 365 nm | 45–20% (150 min) | [47] |
| 6 | PCL | $TiO_2$ NPs (Degussa P-25) | - | Supercritical Foaming & immobilization of $TiO_2$ onto PCL foams | PCL foams with immobilized $TiO_2$ | Acid Orange 7 Basic Yellow 28 | Exposure to an ULTRA-VITALUX lamp 300 W, simulated sunlight irradiation | 100% (24 h) 100% (24 h) | [48] |

Abbreviations: PCL, polycaprolactone; REC, rectorite.

In another interesting study, electrospun PCL/TiO$_2$ fibers were synthesized and were further treated by oxygen plasma method, for photocatalytic purposes. In general, the atmospheric plasma with oxygen gas (OAP) treatment is considered to be a favorable approach to degrade the polymer at the surface of organic–inorganic composites and enhance the hydrophilicity of the fibrous mats. However, as the OAP process partly provoked degradation of PCL throughout the synthetic pathway, OAP-treated samples exhibited deteriorated tensile strength in contrast to the relative untreated one [11]. According to the results of the aforementioned studies, as the TiO$_2$ content increased in the composite photocatalysts, the surface roughness, reinforcement and photocatalytic efficiency of the nanofibers increased, too. However, this photocatalytic efficiency enhancement was only noticed up to a 3 wt% of TiO$_2$ content, since at higher values a deterioration on the mechanical properties of the photocatalyst was observed due to the TiO$_2$ agglomeration.

Solvent-cast processes were also discovered for the preparation of biodegradable PCL photocatalysts with directly immobilized nano-sized TiO$_2$. According to Sökmen et al. [5], two separate solvent casting methodologies were tried: the first included the direct addition of TiO$_2$ dispersion into a PCL solution in dissolving and casting chloroform and the second contained the direct spraying of titania particles/chloroform onto partially solid PCL substrate. Both of the manufactured membranes exhibited biodegradability and self-cleaning properties, whereas the second synthetical route fabricated a more photocatalytic efficient with a 94.2% removal percentage for MB dye. As an expansion of the aforementioned work, Sivlim et al. [47] followed the previous experimental part for the removal of 4-chlorophenol, which was finally more effective in the presence of higher polymer content; a removal attributed to both adsorption and photo-induced degradation of the target compounds.

Another view of the subject provided Marković and coworkers [48], who fabricated floating photocatalysts structured by PCL foam. The synthetic route depended again on two parts with the first being the transformation of PCL beads to foam with the aid of supercritical CO$_2$ (high-pressure cell) and the second the immobilization of TiO$_2$ nanoparticles onto the polymeric foams. The PCL/TiO$_2$ foam exhibited superior photocatalytic performance to the dyes removal than the non-floating one, mainly attributed to the increased surface area and the appearance of pores covered with inorganic particles, facilitating the photo-induced degradation step.

Photocatalytic Performance

According to the available bibliography, PCL-TiO$_2$ composites were applied for the degradation of organic dyes including MB, RhB, and Reactive Black 5 or even for disinfection from bacteria often found in wastewater effluent, such as *E. Coli*, *C. albicans*, and *Staphylococcus aureus*. The PCL composites were manufactured in different structures for photocatalytic purposes, including nanofibers and membranes. Most experiments were carried out under UV light radiation with treatment time ranging from 80–300 min according to the targeted pollutants and the photocatalytic material used. Moreover, the reusability of materials was monitored up to three cycles exhibiting satisfactory results [5,11,12,46–48].

Hu Tu et al. examined the degradation of RhB using PCL/TiO$_2$ nanofibrous catalysts under UV light radiation. The polymeric material enhanced the degradation of RhB due to the advanced structure of PCL mats carriers. Particularly, the obtained results were achieved due to high surface area and porous structure of PCL/TiO$_2$ nanofibrous mats. With the addition of rectorite, a reduction of fibers size was achieved, and the photocatalytic activity was enhanced [12]. In another study, the researchers investigated the degradation efficiency of PCL/TiO$_2$-Ag against different pollutants and bacteria. The highest performance was achieved by the nanofibers with the highest concentration of TiO$_2$. Moreover, the addition of Ag nanoparticles succeeded in an increase of the photocatalytic reaction rate and thus to the enhancement of the overall photocatalytic activity of the composites due to a synergetic effect of Ag nanoparticles and TiO$_2$ catalyst [46]. In another notable study, Darka Markovic et al. investigated the photocatalytic activity of PCL foam doped

with TiO$_2$ nanoparticles, against two azo dyes. The photocatalytic manufactured materials were prepared as floating (PCL$_f$) and non-floating photocatalysts (PCL$_b$). According to their photocatalytic activity, the PCL$_f$/TiO$_2$ had a great performance after three cycles of experiments and better than PCL$_b$, owing to the ability of PCL$_f$ to float on the water, leading to the increase of the UV light adsorption by its surface and as a result an increase of hydroxyl radicals production [48].

### 2.2.3. Other Synthetic Polymers

There are also few biodegradable polymers which have been explored to prepare photocatalysts for TiO$_2$ immobilization, but the relative reports are limited. However, we considered their inclusion within this short review essential as to inspire their further exploration (Table 6). One of them is poly(lactide-co-glycolide) (PLGA), which is a copolymer of poly(lactic acid) (PLA) and poly(glycolic acid) (PGA). It is a biodegradable and biocompatible polyester with tunable mechanical properties; merits that led Pelaseyed et al. [49] to utilize it for the fabrication of 3D porous PLGA/TiO$_2$ nanocomposite scaffolds. Air-liquid foaming technique was employed to manufacture the very porous nanocomposite scaffolds with the PLGA/10 wt% TiO$_2$ being the optimal product, whereas a high photocatalytic efficiency against methylene blue dye was also confirmed, amongst the other beneficial properties of the final composite scaffolds. For the synthesis of a as called "bio-composite" film, another team combined three different biodegradable polymers, namely poly(lactic acid) (PLA), poly(butylene adipate-co-terephthalate) (PBAT) and poly(butylene succinate) (PBS), employing a blown film method (twin screw extruder) to disperse TiO$_2$ nanoparticles in the polymeric matrix. The new eco-friendly materials exhibited good photocatalytic performance for volatile organic compounds' (VOCs) removal [7].

Poly(3-hydroxybutyrate) (PHB) is another interesting biodegradable polymer which is easily degraded by numerous microorganisms (bacteria, fungi, algae) under several conditions. With an innovative concept, PHBcombined with CS oligomers were utilized for the fabrication of fibrous photocatalysts with TiO$_2$ nanoparticles incorporated. Researchers used a grouping of electrospinning, electrospraying and impregnation methods, which potentially ensure the desired architecture of the fibrous scaffolds [13]. The aim of CS oligomers' incorporation into the composite materials is its dynamic stabilization effect of the nano-TiO$_2$ dispersions and their attachment onto the PHB fibers throughout the electrospraying. Due to their excellent photocatalytic performance and reusability, these hybrid and fibrous materials are proposed as favorable candidates for water and air purification from several pollutants.

Regarding PLGA/TiO$_2$ nanocomposite scaffolds, these were applied for the removal of methylene blue dye and against *E. Coli* bacteria with astonishing results. In more details, three different scaffolds were manufactured with different pore sizes. The scaffolds with the smaller pore size, had larger surface area and maximum mechanical strength, leading to an enhanced photocatalytic activity. Moreover, the high photocatalytic performance was attributed to the excellent dispersion of photocatalyst onto the polymeric substrate. Regarding antibacterial activity, the interaction among water, TiO$_2$, and oxygen produced reactive oxygen species. Bacteria released to oxidative stress and were led to death [49].

**Table 6.** Summary of other synthetic polymer-supported TiO$_2$ photocatalysts enclosed in the presented literature.

| | Biodegradable Polymeric Matrix | | | | | Photocatalysis Parameters | | | Ref. |
|---|---|---|---|---|---|---|---|---|---|
| No. | Polymer Substrate | TiO$_2$ Precursor | Dopant | Immobilization technique | Morphology of the Photocatalyst | Type of (Target) Pollutant | Light Source | Degradation Efficiency (Time Required) | |
| 1 | PLA + PBAT + PBS | Titanium isopropoxide (97 wt%) | - | - Sol-gel method for theTiO$_2$ nanoparticles<br>- blown film technique | Composite films | Toluene | Photocatalytic oxidation reactor with UV-C lamp 6 W and λ = 254 nm | 52% (270 min) | [7] |
| 2 | PLGA | TiO$_2$ nanopowder | - | Air-liquid foaming technique | Porous 3D-PCL scaffolds | Methylene blue *E. Coli* | UV lamp light with wavelength 365 nm | 90% (180 min) ~99% (24 h) | [49] |
| 3 | PHB & CS oligomers | Titanium (IV) oxide (nano- TiO$_2$) (99.7% anatase nanopowder) | - | Electrospinning/Electrospraying & Impregnation techniques | Hybrid fibrous materials | Methylene Blue *Escherichia Coli* | UV light (UVASPOT 400/T, Dr. Honle AG; UV lamp UV 400 F/2; 400 W) | >92% (180 min) 100% (30–60 min) | [13] |

Abbreviations: CS, chitosan; PBAT, polybutylene adipate-co-terephthalate; PBS, poly(butylene succinate); PLA, poly(lactic acid); PLGA, poly(lactide-co-glycolide); PHB, poly(3-hydroxybutyrate).

## 3. Conclusions

TiO$_2$-induced photocatalysis considerably remains as the most efficient and feasible option for the photo-degradation of persistent organic pollutants, including mainly pharmaceuticals, azo dyes, toxic metals and pathogenic microorganisms present in water and wastewater. Constant efforts are performing to modify the TiO$_2$ photocatalyst and provide materials highly effective in visible light in accordance with their ease post-treatment recovery. Several research articles have been published in which biodegradable polymers were facilitated to manufacture eco-friendly and sufficient photocatalytic materials against several target pollutants, especially from wastewater. Several methods were applied like sol-gel, film casting, electrospinning, spin coating and 3D printing, transfusing photocatalytic efficiency, mechanical strength and reusability to the fabricated composites. Fibers, membranes and aerogels fabricated from biodegradable polymers, presented different advantages in their overall performance. In fact, the pollutants' removal can be fulfilled by the synergistic effects of the biodegradable polymer-based adsorption and the redox reactions induced by the photo-generated charge carriers, created on the surface of TiO$_2$. A brief presentation of the most novel biodegradable polymer-supported TiO$_2$ photocatalysts, concerning their fabrication pathway and photocatalytic activity, is demonstrated.

Future studies should be focusing on new methodologies and combined techniques for the application of biodegradable polymers, as to prepare chemical and thermal resistant polymer-supported/TiO$_2$ composite materials, with advanced architecture and superior reusability, recyclability and photocatalytic performance. Moreover, the facilitation of green practices for the preparation of the photocatalysts should also be taken into consideration. More research and work are still needed for the categorization of the appropriate manufacturing and TiO$_2$ anchoring techniques for each biodegradable polymer, since each of them possesses a special character with specific chemical and physical properties.

Despite the fact that exceptional studies are found in the literature including the preparation and photocatalytic activity of several polymer-based/TiO$_2$ materials for remediation of wastewater, these works are performed in the laboratory environment and not at large scale. Biodegradable polymer-supported TiO$_2$ materials should be further researched and advanced, exceptionally in the visible light region. Thus, a lot of effort should also be extended for the commercialization of the prepared photocatalysts, under real conditions.

**Author Contributions:** Methodology, N.M.A., D.K., E.E., D.N.B. and D.A.L.; writing—original draft preparation, N.M.A., D.K. and E.E.; writing—review and editing, E.E., D.N.B. and D.A.L.; supervision, D.N.B. and D.A.L. All authors have read and agreed to the published version of the manuscript.

**Funding:** This research has been co-financed by the European Union and Greek national funds through the Operational Program Competitiveness, Entrepreneurship and Innovation, under the call RESEARCH–CREATE–INNOVATE (project code: T2EDK-00137).

**Institutional Review Board Statement:** Not applicable.

**Informed Consent Statement:** Not applicable.

**Data Availability Statement:** The data presented in this study are available on request from the corresponding author.

**Conflicts of Interest:** The authors declare no conflict of interest.

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
