# Peer review of "Insights into Biodegradable Polymer-Supported Titanium Dioxide Photocatalysts for Environmental Remediation"

_2673-6209, doi:10.3390/macromol1030015_

Round 1
Reviewer 1 Report
In this review, the authors reported recent development of TiO2-immobilized materials supported on biodegradable polymers such as chitosan, cellulose, alginate, polyl(actic acid), and polycaprolactone for the photo-degradation of various pollutants.
Bio-degradable polymer-supported TiO2 photocatalysts in this review are promising for environmental application.
Although the condition and mechanism of the reported catalysts is complicated, the characteristic of each catalyst is described carefully.
I think that this review is useful for new researchers to develop the field of polymer-based/TiO2 photocatalyst.
There are some minor comments as follows.
- Table 1, No. 11
sunction filtration ?
- Line 165
TiO2/ZnO2 → TiO2/ZrO2
- Fig. 2, Line 659, Fig. 8, Line 782
TiO2 → TiO2
- Line 479
swift ?
- Line 54
lactid acid → lactic acid
Reviewer 2 Report
The current manuscript provides a comprehensive account of biodegradable polymer-supported TiO2 photocatalysts for environmental remediation. The manuscript is well compiled and well written and I recommend following revisions to further improve the manuscript:
1. The major concern here is too much is discussed on very few polymers. For example, in line with the data presented in Table 1; it is suggested that the authors provide a brief account of their view on the composite systems with multiple polymers, most importantly the natural-synthetic combinations as they are the most commonly used and most efficient in performance. Similarly, the synthetic systems are more reproducible and preferable and only one polymer is discussed there.
2. The degradation profiles of few polymer composite systems should have been provided.
3. Schematics showing the immobilization interaction profile of TiO2 within each of these polymeric systems should have been provided.
Reviewer 3 Report
The review entitedl Novel insights into biodegradable polymer-supported titanium dioxide photocatalysts for environmental remediation is very well written and is worth contrbution. Below are few suggestions that can improve the draft further
1: What is meant by novel in the title. I think this word should be removed as the authors are not presenting or analyzing their point of view. It is just a collection of available literature on the subject.
2: Table 1 is too long. Instead the authors can split it into few tables each one being representing a particular subject. For example the section discussion about photocatalysis should contain the summary of materials used for photocatalysis in the form of Table and so on.
3; The authors have cited 39 references. They should look for all publishing houses such as MDPI, Elsewhere, Wiley, Springer etc so they can claim that they have collected the literature on the subject available during the last decade.
